# Minorization-Maximization for Learning Determinantal Point Processes

**Takahiro Kawashima**                                                          *tkawa@ism.ac.jp*
*Department of Statistical Science*
*The Graduate University for Advanced Studies, SOKENDAI*

**Hideitsu Hino**                                                               *hino@ism.ac.jp*
*The Institute of Statistical Mathematics*
*RIKEN Center for Advanced Intelligence Project*

**Reviewed on OpenReview:** *https://openreview.net/forum?id=65AzNuY73Q*

## Abstract

A determinantal point process (DPP) is a powerful probabilistic model that generates diverse random subsets from a ground set. Since a DPP is characterized by a positive definite kernel, a DPP on a finite ground set can be parameterized by a kernel matrix. Recently, DPPs have gained attention in the machine learning community and have been applied to various practical problems; however, there is still room for further research on the learning of DPPs. In this paper, we propose a simple learning rule for full-rank DPPs based on a minorization-maximization (MM) algorithm, which monotonically increases the likelihood in each iteration. We show that our minorizer of the MM algorithm provides a tighter lower-bound compared to an existing method locally. We also generalize the algorithm for further acceleration. In our experiments on both synthetic and real-world datasets, our method outperforms existing methods in most settings. Our code is available at `https://github.com/ISMHinoLab/DPPMMEstimation`.

## 1 Introduction

A determinantal point process (DPP) is a probabilistic model that represents the occurrence probability of random subsets of a ground set. Initially, DPPs were originated in statistical mechanics to describe the probabilistic behavior of fermions (Macchi, 1975). In recent years, broader applications of DPPs have been developed in the machine learning community (Kulesza & Taskar, 2012).

An important feature of DPPs is the presence of negative dependence (Borcea et al., 2009). There exists some characterizations of negative dependence (Mariet, 2019), and here we consider (pairwise) negative correlation as an example. Letting $\mathcal{A}$ be a random subset, $P(\{i, j\} \subseteq \mathcal{A}) \leq P(i \in \mathcal{A})P(j \in \mathcal{A})$ holds for any pair of items $i, j$ in a ground set when $P(\cdot)$ is defined as a DPP. This means that DPPs can take into account inter-element repulsion, which encourages the occurrence of diverse subsets. This feature aligns with a variety of machine learning applications, such as diversity-promoting image search (Kulesza & Taskar, 2011), recommender systems (Gillenwater et al., 2014), base station configuration for cellular networks (Miyoshi & Shirai, 2014), random design regression (Dereziński et al., 2022), and locating inducing points of sparse variational Gaussian process regression (Burt et al., 2020).

A natural problem on DPPs is efficient learning of the parameters. Since a DPP defined on a finite ground set is parameterized by a positive semidefinite kernel matrix, the learning methods are roughly classified into three approaches: (a) assuming the kernel matrix is full-rank and having no additional structure (full-rank DPPs), (b) assuming the kernel matrix is low-rank (low-rank DPPs), or (c) assuming other tractable structure for the kernel matrix.

So far, some learning methods have been designed for full-rank DPPs. Gillenwater et al. (2014) pioneered the learning problem of DPPs; they developed an EM algorithm for full-rank DPPs. Mariet & Sra (2015) later proposed a fixed-point algorithm for full-rank DPPs. They derived a simple update rule for the kernel matrix and showed its monotonicity by finding its equivalence with a minorization-maximization (MM) algorithm. Their experiments also showed that the fixed-point algorithm is more efficient and stable than the EM algorithm.

Gartrell et al. (2017) introduced low-rank DPPs. Learning of low-rank DPPs involves gradient-based optimization. Mariet et al. (2019) proposed contrastive estimation as an alternative of the maximum likelihood estimation (MLE), while Osogami et al. (2018) incorporated temporal dynamics into low-rank DPPs. A Bayesian extension of low-rank DPPs was also proposed in (Gartrell et al., 2016).

In principle, without special structures, it is difficult to overcome the $\mathcal{O}(N^3)$ time complexity for full-rank DPPs and $\mathcal{O}(NK^2)$ for low-rank DPPs, where $N$ is the size of the ground set and $K$ is the rank of the kernel matrix. To go beyond these complexities, DPPs with special structure are developed, such as Kronecker DPPs (Mariet & Sra, 2016) and the "diagonal+special low-rank" structure (Dupuy & Bach, 2018).

Our study focuses on learning of full-rank DPPs. While full-rank DPPs are sometimes not suitable for problems with a large ground set, we often want to conduct an exact inference for small to medium-sized problems. For example, consider a hypothetical application of a DPP. The first step in the data analysis is to assess whether DPP-based modeling is appropriate for our task or not. Even if our final goal is to handle large data, we typically take relatively small data collected provisionally during this assessment phase. In such a situation, we hope to utilize a ready-made learning algorithm: requiring less hyperparameter tuning, easily implementable, well-behaved, and good convergence speed. However, the existing methods for full-rank DPPs have some difficulties; the EM algorithm (Gillenwater et al., 2014) internally requires optimization on a Stiefel manifold, making the learning procedure complicated and unstable. In (Mariet & Sra, 2015), the authors introduced a step size in order to accelerate the fixed-point algorithm, but the step size was fixed throughout the learning.

In this paper, we propose a simple yet powerful learning rule for full-rank DPPs based on the MM algorithm. Our method increases the log-likelihood monotonically and stably, and locally provides a tighter minorizer than the fixed-point algorithm. Our minorizer is concave while the fixed-point algorithm maximizes a non-concave minorizer in the iteration. This means it has no concern about optimization failure in each iteration. Moreover, we also develop an accelerated version of the proposed MM algorithm. Although the accelerated algorithm requires fixed hyperparameters, the step size is determined adaptively in each iteration. We conduct experiments with both synthetic and real-world datasets and our method outperforms the existing methods in most settings.

In summary, our main contributions in this paper are:

- We present an easy-to-implement learning method for full-rank DPPs based on the MM algorithm. By the property of MM algorithms, our method monotonically increases the log-likelihood.

- We compare the tightness of the minorizers between the existing and proposed methods. The fixed-point algorithm for DPPs proposed in (Mariet & Sra, 2015) can also be viewed as an MM algorithm. Our result indicate that our minorizer locally provides a tighter lower-bound than the existing method. Moreover, our method provides a concave minorizer unlike the exsiting method.

- We derive a generalized form of the minorizer and develop an accelerated algorithm. We also provides an adaptive method to determine the step size values in the iterations for the accelerated algorithm.

- We conduct experiments to evaluate learning algorithms for full-rank DPPs using both synthetic and real-world datasets. Our empirical results show superiority of our method in convergence speed and stability.

## 2 Determinantal Point Processes

Let $k(\cdot,\cdot)$ be a kernel function on a ground set $\Omega$. A determinantal point process (DPP) with the kernel function $k(\cdot,\cdot)$ is a point process on $\Omega$ whose joint intensities are formed as

$$\rho_n(x_1, x_2, \ldots, x_n) = \det(\boldsymbol{K}_{[n]}),$$

where $x_1, x_2, \ldots, x_n \in \Omega$ and $\boldsymbol{K}_{[n]} = (k(x_i, x_j))_{i,j=1}^n$ (Hough et al., 2009). The following theorem gives a sufficient condition for the existence and uniqueness of DPP:

**Theorem 2.1** (Soshnikov (2000); Shirai & Takahashi (2000))**.** *Let $\mathcal{K}$ be a self-adjoint integral operator determined by a kernel function $k$ and be of locally trace class. Then, the kernel function $k(\cdot,\cdot)$ determines a DPP if and only if all the eigenvalues of $\mathcal{K}$ are in $[0,1]$.*

If the restriction of an operator $\mathcal{K}$ to an arbitrary compact subset of $\Omega$ is of trace class, $\mathcal{K}$ is said to be locally trace class. Roughly speaking, Theorem 2.1 states that a positive definite kernel $k(\cdot,\cdot)$ defines a DPP under appropriate scaling which ensures the resulting probabilities in $[0,1]$.

In the context of machine learning, DPPs on a finite ground set $\mathcal{Y} = \{1, 2, \ldots, N\}$ are typically considered. On the finite ground set $\mathcal{Y}$, a point process $P(\cdot)$ is a DPP with a kernel matrix $\boldsymbol{K} \in \mathbb{S}_+^N$ if

$$P(\mathcal{S} \subseteq \mathcal{A}) = \det([\boldsymbol{K}]_{\mathcal{S}})$$

for a random subset $\mathcal{A} \subseteq \mathcal{Y}$ drawn by $P$ and an arbitrary $\mathcal{S} \subseteq \mathcal{Y}$. $[\boldsymbol{K}]_{\mathcal{S}} = (K_{ij})_{i,j \in \mathcal{S}} \in \mathbb{S}_+^{|\mathcal{S}|}$ denotes the principal submatrix of $\boldsymbol{K}$ and the kernel matrix $\boldsymbol{K}$ must be $\boldsymbol{O} \preceq \boldsymbol{K} \preceq \boldsymbol{I}$ from an analogy with the DPPs on a general ground set[1]. A DPP on a finite ground set has an alternative representation called the $\boldsymbol{L}$-ensemble (Borodin & Rains, 2005), which defines the occurrence probability of a random subset $\mathcal{A} \subseteq \mathcal{Y}$ as

$$P_{\boldsymbol{L}}(\mathcal{A}) = \frac{\det([\boldsymbol{L}]_{\mathcal{A}})}{\det(\boldsymbol{L} + \boldsymbol{I})},$$

where $\boldsymbol{L} \in \mathbb{S}_+^N$ is a positive semidefinite kernel matrix. We can commute between $\boldsymbol{K}$ and $\boldsymbol{L}$ using the equation $\boldsymbol{K} = \boldsymbol{L}(\boldsymbol{L}+\boldsymbol{I})^{-1}$ or its inversion $\boldsymbol{L} = \boldsymbol{K}(\boldsymbol{I}-\boldsymbol{K})^{-1}$ if $\boldsymbol{I} - \boldsymbol{K}$ is invertible. In this paper, we develop a learning algorithm for $\boldsymbol{L}$.

## 3 Learning Algorithm

Given $M$ samples $\mathcal{A}_1, \mathcal{A}_2, \ldots, \mathcal{A}_M \subseteq \mathcal{Y}$, our goal is to solve MLE. That is, to find a maximizer of the log-likelihood

$$\begin{aligned} f(\boldsymbol{L}) &= \frac{1}{M} \sum_{m=1}^M \log \det([\boldsymbol{L}]_{\mathcal{A}_m}) - \log \det(\boldsymbol{L} + \boldsymbol{I}) \\ &= \frac{1}{M} \sum_{m=1}^M \log \det(\boldsymbol{U}_{\mathcal{A}_m} \boldsymbol{L} \boldsymbol{U}_{\mathcal{A}_m}^\top) - \log \det(\boldsymbol{L} + \boldsymbol{I}), \end{aligned} \tag{1}$$

where $\boldsymbol{U}_{\mathcal{A}_m} \in \{0,1\}^{|\mathcal{A}_m| \times N}$ is the submatrix of $\boldsymbol{I}$ obtained by keeping the rows corresponding to the elements in $\mathcal{A}_m$.

### 3.1 MM Algorithm

A minorization-maximization (MM) algorithm is a powerful meta-algorithm for finding a local maximizer of a generally non-concave objective $f(\theta)$ (Hunter & Lange, 2004; Sun et al., 2017). The MM algorithm consists of two steps: (i) find a minorizer $g(\theta|\theta^{(t)})$ of $f(\theta)$ that satisfies

---

[1]We use $\prec, \preceq, \succ,$ and $\succeq$ in the sense of positive (semi-)definite ordering.

- $f(\theta) \geq g(\theta|\theta^{(t)})$

- $f(\theta^{(t)}) = g(\theta^{(t)}|\theta^{(t)})$

for all $\theta$ and $\theta^{(t)}$ within a feasible region. Then, (ii) maximize the minorizer $g(\theta|\theta^{(t)})$ with respect to $\theta$ and set $\theta^{(t+1)} = \arg \max g(\theta|\theta^{(t)})$. Repeating this process, we can obtain a sequence of the parameters $\{\theta^{(t)}\}_{t \geq 0}$ which monotonically increases the objective value, because

$$f(\theta^{(t+1)}) \geq g(\theta^{(t+1)}|\theta^{(t)}) \geq g(\theta^{(t)}|\theta^{(t)}) = f(\theta^{(t)}) \tag{2}$$

holds.

Since $\log \det(\cdot)$ is concave on $\mathbb{S}_{++}$, the objective function (1) is a combination of concave and convex functions. From the concavity of $\log \det(\cdot)$, the following linear upper bound is derived with the first-order Taylor expansion

$$\log \det(\boldsymbol{X}) \leq \log \det(\boldsymbol{Y}) + \mathrm{tr}\{\boldsymbol{Y}^{-1}(\boldsymbol{X} - \boldsymbol{Y})\} = \log \det(\boldsymbol{Y}) + \mathrm{tr}(\boldsymbol{Y}^{-1}\boldsymbol{X}) - n \tag{3}$$

for any $\boldsymbol{X}, \boldsymbol{Y} \in \mathbb{S}_{++}^n, n \in \mathbb{N}$, and by swapping $\boldsymbol{X}$ and $\boldsymbol{Y}$,

$$\log \det(\boldsymbol{X}) \geq \log \det(\boldsymbol{Y}) - \mathrm{tr}\{\boldsymbol{X}^{-1}(\boldsymbol{Y} - \boldsymbol{X})\} = \log \det(\boldsymbol{Y}) - \mathrm{tr}(\boldsymbol{X}^{-1}\boldsymbol{Y}) + n \tag{4}$$

also holds. From (3) with $\boldsymbol{X} \to \boldsymbol{L} + \boldsymbol{I}$ and $\boldsymbol{Y} \to \boldsymbol{L}^{(t+1)} + \boldsymbol{I}$, we have

$$-\log \det(\boldsymbol{L} + \boldsymbol{I}) \geq -\log \det(\boldsymbol{L}^{(t)} + \boldsymbol{I}) - \mathrm{tr}\{(\boldsymbol{L}^{(t)} + \boldsymbol{I})^{-1}(\boldsymbol{L} - \boldsymbol{L}^{(t)})\}, \tag{5}$$

which yields a choice for minorizing the objective (1). This method is referred to as the concave-convex procedure (CCCP) (Yuille & Rangarajan, 2001), a special case of MM algorithms. However, the minorizer derived by the CCCP has no closed-form maximizer in our case, therefore, we devise an easy-to-optimize alternative.

## 3.2 Proposed Algorithm

In the proposed minorizer of (1), the convex part is lower-bounded linearly by (5) and the concave part $\log \det([\boldsymbol{L}]_{\mathcal{A}_m}) = \log \det(\boldsymbol{U}_{\mathcal{A}_m} \boldsymbol{L} \boldsymbol{U}_{\mathcal{A}_m}^\top)$ is also lower-bounded. The following proposition provides the concrete form of our proposed minorizer.

**Proposition 3.1.** *Let $f(\boldsymbol{L})$ be given by (1) and*

$$g(\boldsymbol{L}|\boldsymbol{L}^{(t)}) = -\frac{1}{M} \sum_{m=1}^{M} \mathrm{tr}\{\boldsymbol{L}^{(t)} \boldsymbol{U}_{\mathcal{A}_m}^\top [\boldsymbol{L}^{(t)}]_{\mathcal{A}_m}^{-1} \boldsymbol{U}_{\mathcal{A}_m} \boldsymbol{L}^{(t)} \boldsymbol{L}^{-1}\} - \mathrm{tr}\{(\boldsymbol{L}^{(t)} + \boldsymbol{I})^{-1} \boldsymbol{L}\} + \zeta(\boldsymbol{L}^{(t)}), \tag{6}$$

*where*

$$\zeta(\boldsymbol{L}^{(t)}) = \frac{1}{M} \sum_{m=1}^{M} \left\{\log \det(\boldsymbol{U}_{\mathcal{A}_m} \boldsymbol{L}^{(t)} \boldsymbol{U}_{\mathcal{A}_m}^\top) + |\mathcal{A}_m|\right\} - \log \det(\boldsymbol{L}^{(t)} + \boldsymbol{I}) + \mathrm{tr}\{(\boldsymbol{L}^{(t)} + \boldsymbol{I})^{-1} \boldsymbol{L}^{(t)}\}$$

*is a constant term. Then, $f(\boldsymbol{L}) \geq g(\boldsymbol{L}|\boldsymbol{L}^{(t)})$ and $f(\boldsymbol{L}^{(t)}) = g(\boldsymbol{L}^{(t)}|\boldsymbol{L}^{(t)})$ hold for any $\boldsymbol{L}, \boldsymbol{L}^{(t)} \in \mathbb{S}_{++}^N$.*

*Proof.* For any positive definite $\boldsymbol{P}, \boldsymbol{P}_t \succ 0$ and any square or tall non-generated matrix $\boldsymbol{A}$, the following matrix inequality holds (Sun et al., 2016; 2017):

$$(\boldsymbol{A}\boldsymbol{P}\boldsymbol{A}^\top)^{-1} \preceq \boldsymbol{R}_t^{-1} \boldsymbol{A} \boldsymbol{P}_t \boldsymbol{P}^{-1} \boldsymbol{P}_t \boldsymbol{A}^\top \boldsymbol{R}_t^{-1},$$
$$\boldsymbol{R}_t = \boldsymbol{A} \boldsymbol{P}_t \boldsymbol{A}^\top,$$

and thus we have

$$\mathrm{tr}\{(\boldsymbol{A}\boldsymbol{P}\boldsymbol{A}^\top)^{-1} \boldsymbol{S}\} \leq \mathrm{tr}\{\boldsymbol{R}_t^{-1} \boldsymbol{A} \boldsymbol{P}_t \boldsymbol{P}^{-1} \boldsymbol{P}_t \boldsymbol{A}^\top \boldsymbol{R}_t^{-1} \boldsymbol{S}\} \tag{7}$$

---

**Algorithm 1:** Minorization-Maximization (MM)

---

**Input:** Training set $\{\mathcal{A}_1, \mathcal{A}_2, \ldots, \mathcal{A}_M\}$, initial value $\boldsymbol{L} \succ \boldsymbol{O}$, and machine epsilon $\varepsilon \geq 0$
**Output:** $\boldsymbol{L}$
**for** $t = 1$ *to* $T$ **do**

$\quad \boldsymbol{A} \leftarrow \boldsymbol{O}$;

$\quad \boldsymbol{Q}_\varepsilon \leftarrow \boldsymbol{L} \left( \dfrac{1}{M} \displaystyle\sum_{m=1}^{M} \boldsymbol{U}_{\mathcal{A}_m}^\top [\boldsymbol{L}]_{\mathcal{A}_m}^{-1} \boldsymbol{U}_{\mathcal{A}_m} \right) \boldsymbol{L} + \varepsilon \boldsymbol{I}$;

$\quad \boldsymbol{G} \leftarrow (\boldsymbol{L} + \boldsymbol{I})^{-1}$;

$\quad \boldsymbol{L} \leftarrow \text{SolveCARE}(\boldsymbol{A}, \boldsymbol{Q}_\varepsilon, \boldsymbol{G})$; // Solve Equation (12)

**end**

---

for any appropriately sized and positive semidefinite $\boldsymbol{S} \succeq \boldsymbol{O}$. Using the lower-bound (4) with the substitutions $\boldsymbol{X} \rightarrow \boldsymbol{U}_{\mathcal{A}_m} \boldsymbol{L} \boldsymbol{U}_{\mathcal{A}_m}^\top$, $\boldsymbol{Y} \rightarrow \boldsymbol{U}_{\mathcal{A}_m} \boldsymbol{L}^{(t)} \boldsymbol{U}_{\mathcal{A}_m}^\top$ and (7) with $\boldsymbol{A} \rightarrow \boldsymbol{U}_{\mathcal{A}_m}, \boldsymbol{P} \rightarrow \boldsymbol{L}, \boldsymbol{P}_t \rightarrow \boldsymbol{L}^{(t)}$ and $\boldsymbol{S} \rightarrow \boldsymbol{U}_{\mathcal{A}_m} \boldsymbol{L}^{(t)} \boldsymbol{U}_{\mathcal{A}_m}^\top$, we have

$$\log \det(\boldsymbol{U}_{\mathcal{A}_m} \boldsymbol{L} \boldsymbol{U}_{\mathcal{A}_m}^\top) \geq |\mathcal{A}_m| + \log \det(\boldsymbol{U}_{\mathcal{A}_m} \boldsymbol{L}^{(t)} \boldsymbol{U}_{\mathcal{A}_m}^\top) - \text{tr}\{(\boldsymbol{U}_{\mathcal{A}_m} \boldsymbol{L} \boldsymbol{U}_{\mathcal{A}_m}^\top)^{-1} \boldsymbol{U}_{\mathcal{A}_m} \boldsymbol{L}^{(t)} \boldsymbol{U}_{\mathcal{A}_m}^\top\}$$
$$\geq |\mathcal{A}_m| + \log \det(\boldsymbol{U}_{\mathcal{A}_m} \boldsymbol{L}^{(t)} \boldsymbol{U}_{\mathcal{A}_m}^\top) - \text{tr}\{\boldsymbol{L}^{(t)} \boldsymbol{U}_{\mathcal{A}_m}^\top [\boldsymbol{L}^{(t)}]_{\mathcal{A}_m}^{-1} \boldsymbol{U}_{\mathcal{A}_m} \boldsymbol{L}^{(t)} \boldsymbol{L}^{-1}\}. \quad (8)$$

Combining the lower-bounds (5) and (8), we can construct the minorizer of $f(\boldsymbol{L})$ as (6). $\qquad \square$

In order to obtain the maximizer of (1), we iteratively optimize the proposed minorizer $g(\boldsymbol{L}|\boldsymbol{L}^{(t)})$ by solving the first-order optimality condition for $t = 1, \ldots, T$. Since $g(\boldsymbol{L}|\boldsymbol{L}^{(t)})$ is concave because of the convexity of $\text{tr}(\boldsymbol{X}^{-1})$ for $\boldsymbol{X} \succ 0$, a stationary point of $g(\boldsymbol{L}|\boldsymbol{L}^{(t)})$ is also its global maximizer.

**Proposition 3.2.** *A global maximizer of $g(\boldsymbol{L}|\boldsymbol{L}^{(t)})$ satisfies*

$$-\boldsymbol{L}(\boldsymbol{L}^{(t)} + \boldsymbol{I})^{-1}\boldsymbol{L} + \boldsymbol{Q}_M^{(t)} = \boldsymbol{O}, \quad (9)$$

*where*

$$\boldsymbol{Q}_M^{(t)} = \boldsymbol{L}^{(t)} \left( \frac{1}{M} \sum_{m=1}^{M} \boldsymbol{U}_{\mathcal{A}_m}^\top [\boldsymbol{L}^{(t)}]_{\mathcal{A}_m}^{-1} \boldsymbol{U}_{\mathcal{A}_m} \right) \boldsymbol{L}^{(t)}. \quad (10)$$

*Proof.* Noting that $\nabla_{\boldsymbol{X}} \text{tr}(\boldsymbol{A}\boldsymbol{X}) = \boldsymbol{A}^\top$ and $\nabla_{\boldsymbol{X}} \text{tr}(\boldsymbol{A}\boldsymbol{X}^{-1}) = -(\boldsymbol{X}^{-1}\boldsymbol{A}\boldsymbol{X}^{-1})^\top$ for appropriate matrices $\boldsymbol{X}$ and $\boldsymbol{A}$, the optimality condition of (6) is

$$\nabla_{\boldsymbol{L}} g(\boldsymbol{L}|\boldsymbol{L}^{(t)}) = \boldsymbol{L}^{-1} \boldsymbol{Q}_M^{(t)} \boldsymbol{L}^{-1} - (\boldsymbol{L}^{(t)} + \boldsymbol{I})^{-1} = \boldsymbol{O}. \quad (11)$$

By multiplying both sides of (11) by $\boldsymbol{L}$, we can see that the stationary points of $g(\boldsymbol{L}|\boldsymbol{L}^{(t)})$ satisfy (9). From the concavity of $g(\boldsymbol{L}|\boldsymbol{L}^{(t)})$, we obtain the result. $\qquad \square$

The matrix quadratic equation (9) is a special case of the continuous algebraic Riccati equation (CARE):

$$\boldsymbol{A}^\top \boldsymbol{X} + \boldsymbol{X}\boldsymbol{A} - \boldsymbol{X}\boldsymbol{G}\boldsymbol{X} + \boldsymbol{Q} = \boldsymbol{O}, \quad (12)$$

where $\boldsymbol{X} \in \mathbb{S}^N$ is unknown, and $\boldsymbol{G}, \boldsymbol{Q} \in \mathbb{S}^N, \boldsymbol{A} \in \mathbb{R}^{N \times N}$ are fixed coefficient matrices. The CARE is well-studied in control engineering and is solvable by some numerical methods such as the Schur method (Laub, 1979) and Newton's method (Bini et al., 2011; Benner & Byers, 1998). It is worth noting that CARE solvers are available in most programming languages through packages for scientific computation; for example, `SciPy` in Python and `MatrixEquations.jl` in Julia.

In addition, we can confirm the following statement as a corollary of Proposition 3.2.

**Corollary 3.1.** *With the same notation as in Proposition 3.2 and a positive definite initial value $\boldsymbol{L}^{(0)} \succ 0$, we have* $\operatorname{rank}(\boldsymbol{L}^{(t)}) = \operatorname{rank}(\boldsymbol{Q}_M^{(t)})$ *for* $t = 1, 2, \ldots$.

*Proof.* Since $\boldsymbol{L}^{(0)}$ is positive definite, $(\boldsymbol{L}^{(0)} + \boldsymbol{I})^{-1}$ is non-singular. Therefore, from the optimality condition (9),

$$\operatorname{rank}(\boldsymbol{L}^{(1)}(\boldsymbol{L}^{(0)} + \boldsymbol{I})^{-1} \boldsymbol{L}^{(1)}) = \operatorname{rank}(\boldsymbol{L}^{(1)}) = \operatorname{rank}(\boldsymbol{Q}_M^{(1)}).$$

By applying similar operations recursively, the result can be confirmed. □

From the assumption in Corollary 3.1, we find that $\boldsymbol{L}^{(0)}$ should be initialized by some positive definite matrix. See the experimental settings described in Section 5 for examples of the initialization. Corollary 3.1 also says that if $\boldsymbol{Q}_M^{(t)}$ is degenerated, the solution of (9) must also be degenerated. This means that when $\boldsymbol{Q}_M^{(t)}$ is singular, the solution of (9) falls outside the feasible region $\mathbb{S}_{++}^N$. This problem arises when some elements of $\mathcal{Y}$ are never observed in the given data $\mathcal{A}_1, \mathcal{A}_2, \ldots, \mathcal{A}_M$, that is, $\bigcup_{m=1}^M \mathcal{A}_m \subsetneq \mathcal{Y}$ holds. To avoid this issue and stabilize numerical computation, we recommend to solve

$$-\boldsymbol{L}(\boldsymbol{L}^{(t)} + \boldsymbol{I})^{-1} \boldsymbol{L} + \boldsymbol{Q}_M^{(t)} + \varepsilon \boldsymbol{I} = \boldsymbol{O}$$

with a machine epsilon $\varepsilon > 0$ instead of (9). We note that the choice of the machine epsilon $\varepsilon$ does not affect the estimate significantly; we use $\varepsilon = 10^{-10}$ throughout this paper. The procedure for the proposed MM-based learning is summarized in Algorithm 1.

### 3.3 Relation to the Existing Method

Mariet & Sra (2015) derived the following update rule to maximize (1) as a fixed-point algorithm:

$$\boldsymbol{L}^{(t+1)} = \boldsymbol{L}^{(t)} + a\boldsymbol{L}^{(t)} \nabla f(\boldsymbol{L}^{(t)}) \boldsymbol{L}^{(t)}, \tag{13}$$

$$\nabla f(\boldsymbol{L}) = \frac{1}{M} \sum_{m=1}^M \boldsymbol{U}_{\mathcal{A}_m}^\top [\boldsymbol{L}]_{\mathcal{A}_m}^{-1} \boldsymbol{U}_{\mathcal{A}_m} - (\boldsymbol{L} + \boldsymbol{I})^{-1},$$

where $a > 0$ is a step size. For $a = 1$, they also show that the update rule (13) can also be regarded as an MM algorithm with the non-concave minorizer

$$h(\boldsymbol{L}|\boldsymbol{L}^{(t)}) = -\frac{1}{M} \sum_{m=1}^M \operatorname{tr}\{\boldsymbol{L}^{(t)} \boldsymbol{U}_{\mathcal{A}_m}^\top [\boldsymbol{L}^{(t)}]_{\mathcal{A}_m}^{-1} \boldsymbol{U}_{\mathcal{A}_m} \boldsymbol{L}^{(t)} \boldsymbol{L}^{-1}\}$$

$$- \log \det(\boldsymbol{L}) - \operatorname{tr}\{(\boldsymbol{L}^{(t)} + \boldsymbol{I})^{-1} \boldsymbol{L}^{-1} \boldsymbol{L}^{(t)}\} + \xi(\boldsymbol{L}^{(t)}), \tag{14}$$

where $\xi(\boldsymbol{L}^{(t)})$ is a constant term and explicitly given in Appendix A. Comparing (14) with (6), we can see that the lower-bounds for the first term in (1) are the same, and those for the second term only differ. With respect to these minorizers, the following proposition holds.

**Proposition 3.3.** *For $g(\boldsymbol{L}|\boldsymbol{L}^{(t)})$ defined in (6) and $h(\boldsymbol{L}|\boldsymbol{L}^{(t)})$ defined in (14), it holds that $g(\boldsymbol{L}|\boldsymbol{L}^{(t)}) \geq h(\boldsymbol{L}|\boldsymbol{L}^{(t)})$ for $\boldsymbol{L}$ in the $\delta$-neighborhood of $\boldsymbol{L}^{(t)}$: $\mathcal{B}_\delta(\boldsymbol{L}^{(t)}) = \{\boldsymbol{L}^{(t)} + \delta \boldsymbol{M} : \boldsymbol{M}$ is a symmetric matrix whose eigenvalues are all in $[-1, 1]\}$ with a sufficiently small $\delta > 0$.*

*Proof.* We have the following inequality:

$$g(\boldsymbol{L}|\boldsymbol{L}^{(t)}) - h(\boldsymbol{L}|\boldsymbol{L}^{(t)}) \geq \operatorname{tr}\{(\boldsymbol{L}^{(t)} + \boldsymbol{I})^{-1}(2\boldsymbol{L}^{(t)} - \boldsymbol{L} - \boldsymbol{L}^{(t)} \boldsymbol{L}^{-1} \boldsymbol{L}^{(t)})\}, \tag{15}$$

where the derivation is shown in Appendix A. If $\boldsymbol{L} \in \mathcal{B}_\delta(\boldsymbol{L}^{(t)})$, we have

$$2\boldsymbol{L}^{(t)} - \boldsymbol{L} - \boldsymbol{L}^{(t)} \boldsymbol{L}^{-1} \boldsymbol{L}^{(t)} \approx \boldsymbol{O}. \tag{16}$$

The details of the derivation can be found in the appendix. Applying the approximation (16) to (15), we can conclude the proposition. □

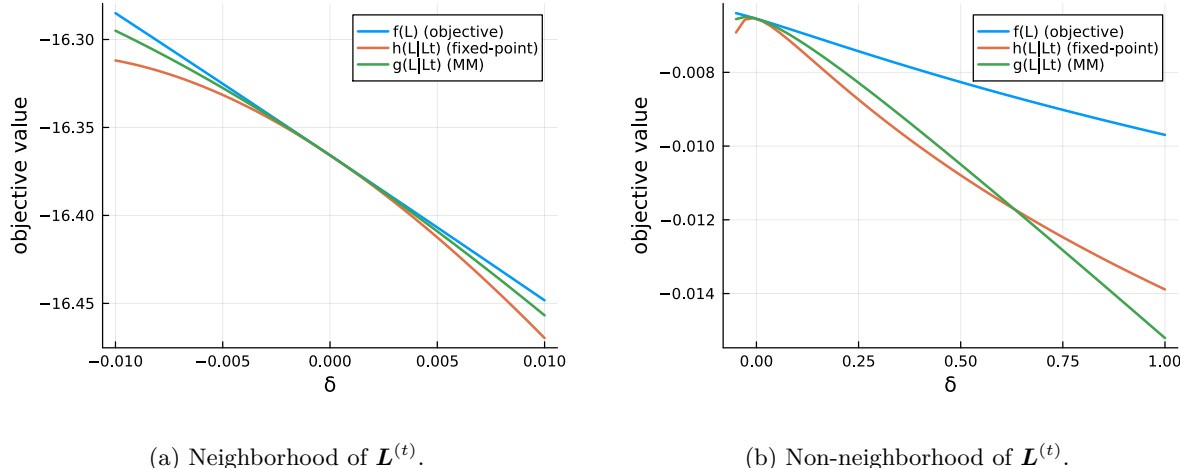

(a) Neighborhood of $\boldsymbol{L}^{(t)}$.

(b) Non-neighborhood of $\boldsymbol{L}^{(t)}$.

Figure 1: Behavior of minorizers.

The proposition 3.3 states that the proposed minorizer gives a tighter lower-bound of the objective than that of the existing method locally. This leads to a tighter leftmost inequality in (2), making it likely that the proposed method will produce better $\boldsymbol{L}^{(t+1)}$. Figure 1 shows the behavior of the minorizers in the neighborhood and non-neighborhood of $\boldsymbol{L}^{(t)}$. The proposed minorizer becomes looser as $\boldsymbol{L}$ moves farther away from $\boldsymbol{L}^{(t)}$, but the experimental results in Section 5 show that the proposed method converges faster in most cases. Note that the minorizer of the fixed-point algorithm is non-convex as seen in Figure 1(b). This implies that the fixed-point algorithm is possible to get trapped in poor stationary points of $h(\boldsymbol{L}|\boldsymbol{L}^{(t)})$.

### 3.4 Computational Costs

In our method, the total computational cost per iteration is $\mathcal{O}(M\kappa^3 + N^3)$, where $\kappa = \max_m |\mathcal{A}_m|$. It is computed as follows; the computation of $\boldsymbol{Q}_M^{(t)}$ in (9) requires $\mathcal{O}(\sum_{m=1}^M |\mathcal{A}_m|^3 + N^3) = \mathcal{O}(M\kappa^3 + N^3)$ operations, including the evaluation of $[\boldsymbol{L}^{(t)}]_{\mathcal{A}_m}^{-1}$ for all $m = 1, 2, \ldots, M$ and the matrix multiplications of $N \times N$ matrices. The inversion $(\boldsymbol{L}^{(t)} + \boldsymbol{I})^{-1}$ and solving the CARE also cost $\mathcal{O}(N^3)$.

The computational complexity of our method is equal to that of the fixed-point algorithm (Mariet & Sra, 2015). Although our method incurs additional $\mathcal{O}(N^3)$ computations due to the CARE, the experimental results in Section 5 show faster convergence of our method in computational time. We note that the gradient-based learning of a low-rank factorized DPP also takes the same $\mathcal{O}(M\kappa^3 + N^3)$ per iteration if the factorization is full-rank (Gartrell et al., 2017; Osogami et al., 2018).

## 4 Generalization and Acceleration

In this section, we develop generalization of the minorizer (6) and the CARE (9) for further acceleration of the algorithm.

### 4.1 Generalizing the Minorizer

By adding a penalty term to the mean log-likelihood (1), we can generalize the objective as

$$f_{\mu^{(t)}}(\boldsymbol{L}|\boldsymbol{L}^{(t)}) = f(\boldsymbol{L}) - \mu^{(t)}d(\boldsymbol{L}\|\boldsymbol{L}^{(t)}), \tag{17}$$

where $\mu^{(t)} \geq 0$ is a non-negative coefficient and $d(\cdot\|\cdot)$ is an appropriate divergence defined on $\mathbb{S}_{++}^N \times \mathbb{S}_{++}^N$. The additional penalty term $\mu^{(t)}d(\boldsymbol{L}\|\boldsymbol{L}^{(t)})$ effects to prevent a big change from $\boldsymbol{L}^{(t)}$ to $\boldsymbol{L}^{(t+1)}$. By the definition

---

**Algorithm 2:** Accelerated MM

---

**Input:** Training set $\{\mathcal{A}_1, \mathcal{A}_2, \ldots, \mathcal{A}_M\}$, initial value $\boldsymbol{L} \succ \boldsymbol{O}$, machine epsilon $\varepsilon \geq 0$, tolerance $\delta \in (0, 1)$,
        and acceleration steps $T_{\mathrm{acc}} \in \{0, 1, \ldots, T\}$

**Output:** $\boldsymbol{L}$

**for** $t = 1$ *to* $T$ **do**

$\quad \boldsymbol{H} \leftarrow \left( \dfrac{1}{M} \displaystyle\sum_{m=1}^{M} \boldsymbol{U}_{\mathcal{A}_m}^{\top} [\boldsymbol{L}]_{\mathcal{A}_m}^{-1} \boldsymbol{U}_{\mathcal{A}_m} \right);$

$\quad$ **if** $t \leq T_{\mathrm{acc}}$ **then**

$\quad\quad \mid \mu \leftarrow \min \{\max \{-1/\lambda_{\max}(\boldsymbol{H}(\boldsymbol{L} + \boldsymbol{I})), -1\} + \delta, 0\};$

$\quad$ **end**

$\quad$ **else**

$\quad\quad \mid \mu \leftarrow 0;$

$\quad$ **end**

$\quad \boldsymbol{A} \leftarrow \boldsymbol{O};$

$\quad \boldsymbol{Q}_{\mu, \varepsilon} \leftarrow (1 + \mu) \boldsymbol{LHL} + \varepsilon \boldsymbol{I};$

$\quad \boldsymbol{G}_{\mu} \leftarrow \mu \boldsymbol{H} + (\boldsymbol{L} + \boldsymbol{I})^{-1};$

$\quad \boldsymbol{L} \leftarrow \mathrm{SolveCARE}(\boldsymbol{A}, \boldsymbol{Q}_{\mu, \varepsilon}, \boldsymbol{G}_{\mu});$ // Solve Equation (12)

**end**

---

of a divergence, $d(\boldsymbol{L} \| \boldsymbol{L}^{(t)}) \geq 0$ for any $\boldsymbol{L}, \boldsymbol{L}^{(t)} \in \mathbb{S}_{++}^N$ and the equality holds if and only if $\boldsymbol{L} = \boldsymbol{L}^{(t)}$. This means that the generalized objective $f_{\mu^{(t)}}(\boldsymbol{L} | \boldsymbol{L}^{(t)})$ also works as the minorizer of $f(\boldsymbol{L})$. Specifically, such a scheme is called as the proximal point algorithm if the divergence $d(\cdot \| \cdot)$ is the squared Euclidean distance (Parikh & Boyd, 2014). Or it is also called mirror ascent (descent) or Bregman minorization (majorization) if $d(\cdot \| \cdot)$ is a Bregman divergence (Nemirovsky, 1983; Beck & Teboulle, 2003; Lange et al., 2021).

In our case, we consider a logdet divergence:

$$D_{\mathrm{ld}}(\boldsymbol{X} \| \boldsymbol{Y}) = -\log \det(\boldsymbol{X}) + \log \det(\boldsymbol{Y}) + \mathrm{tr}\{\boldsymbol{Y}^{-1}(\boldsymbol{X} - \boldsymbol{Y})\},$$

and define $d(\cdot \| \cdot)$ as

$$d(\boldsymbol{L} \| \boldsymbol{L}^{(t)}) = \frac{1}{M} \sum_{m=1}^{M} D_{\mathrm{ld}}([\boldsymbol{L}^{(t)}]_{\mathcal{A}_m} \| [\boldsymbol{L}]_{\mathcal{A}_m}). \tag{18}$$

The defined $d(\cdot \| \cdot)$ in (18) satisfies the definition of a divergence if and only if $\bigcup_m \mathcal{A}_m = \mathcal{Y}$ holds. The divergence (18) leads the following minorizer of $f(\boldsymbol{L})$ and $f_{\mu^{(t)}}(\boldsymbol{L} | \boldsymbol{L}^{(t)})$.

**Proposition 4.1.** *Let $f(\boldsymbol{L})$ be defined in (1) and $f_{\mu^{(t)}}(\boldsymbol{L} | \boldsymbol{L}^{(t)})$ be defined in (17) with $\mu^{(t)} \geq 0$ and the divergence (18). Then, the concave function*

$$g_{\mu^{(t)}}(\boldsymbol{L} | \boldsymbol{L}^{(t)}) = -\frac{1 + \mu^{(t)}}{M} \sum_{m=1}^{M} \mathrm{tr}(\boldsymbol{L}^{(t)} \boldsymbol{U}_{\mathcal{A}_m}^{\top} [\boldsymbol{L}^{(t)}]_{\mathcal{A}_m}^{-1} \boldsymbol{U}_{\mathcal{A}_m} \boldsymbol{L}^{(t)} \boldsymbol{L}^{-1})$$

$$- \frac{\mu^{(t)}}{M} \sum_{m=1}^{M} \mathrm{tr}([\boldsymbol{L}^{(t)}]_{\mathcal{A}_m}^{-1} [\boldsymbol{L}]_{\mathcal{A}_m}) - \mathrm{tr}\{(\boldsymbol{L}^{(t)} + \boldsymbol{I})^{-1} \boldsymbol{L}\} + \zeta_{\mu^{(t)}}(\boldsymbol{L}^{(t)}),$$

*where $\zeta_{\mu^{(t)}}(\boldsymbol{L}^{(t)})$ is a constant term, is the minorizer of $f(\boldsymbol{L})$ and $f_{\mu^{(t)}}(\boldsymbol{L} | \boldsymbol{L}^{(t)})$.*

See Appendix B for the proof.

We can maximize $g_{\mu^{(t)}}(\cdot | \boldsymbol{L}^{(t)})$ by solving a CARE in the same manner as Proposition 3.2.

**Proposition 4.2.** *A global maximizer of $g_{\mu^{(t)}}(\boldsymbol{L} | \boldsymbol{L}^{(t)})$ satisfies the CARE*

$$-\boldsymbol{L} \left\{ \mu^{(t)} \boldsymbol{H}_M^{(t)} + (\boldsymbol{L}^{(t)} + \boldsymbol{I})^{-1} \right\} \boldsymbol{L} + (1 + \mu^{(t)}) \boldsymbol{L}^{(t)} \boldsymbol{H}_M^{(t)} \boldsymbol{L}^{(t)} = \boldsymbol{O}, \tag{19}$$

*where*

$$\boldsymbol{H}_M^{(t)} = \frac{1}{M} \sum_{m=1}^M \boldsymbol{U}_{\mathcal{A}_m}^\top [\boldsymbol{L}^{(t)}]_{\mathcal{A}_m}^{-1} \boldsymbol{U}_{\mathcal{A}_m}.$$

$\boldsymbol{H}_M^{(t)}$ degenerates if $\bigcup_{m=1}^M \mathcal{A}_m \subsetneq \mathcal{Y}$ holds as well as $\boldsymbol{Q}_M^{(t)} = \boldsymbol{L}^{(t)} \boldsymbol{H}_M^{(t)} \boldsymbol{L}^{(t)}$ defined in (10). For $\mu^{(t)} = 0$, we have $g_{\mu^{(t)}}(\boldsymbol{L}|\boldsymbol{L}^{(t)}) = g(\boldsymbol{L}|\boldsymbol{L}^{(t)})$ and the update rule (19) comes down to the original CARE (9). For $\mu^{(t)} > 0$, the update rule (19) also works as the MM iteration but the convergence may become slower by the penalty term.

### 4.2 Acceleration and Hyperparameter Determination

What happens if the coefficient $\mu^{(t)}$ is set to negative? Then, $f_{\mu^{(t)}}(\boldsymbol{L}|\boldsymbol{L}^{(t)})$ and $g_{\mu^{(t)}}(\boldsymbol{L}|\boldsymbol{L}^{(t)})$ can no longer be regarded as the minorizers, but it is expected that the update rule produces a bigger change from $\boldsymbol{L}^{(t)}$ to $\boldsymbol{L}^{(t+1)}$ and the learning speed may become faster. However, similar to the learning rate of a gradient descent, a too large absolute value for $\mu^{(t)} < 0$ may lead bad convergence. What is worse, the solution of the CARE (19) can even not exist. Our approach to decide the negative $\mu^{(t)} < 0$ is to ensure that there is at least a feasible solution to the CARE (19).

**Lemma 4.1.** *Let $\boldsymbol{G} \in \mathbb{S}^N$ and $\boldsymbol{Q} \in \mathbb{S}_{++}^N$ be fixed coefficients and $\boldsymbol{X} \in \mathbb{S}^N$ be unknown. Then, the following equation*

$$\boldsymbol{X}\boldsymbol{G}\boldsymbol{X} = \boldsymbol{Q} \tag{20}$$

*has a solution in $\mathbb{S}_{++}^N$ if and only if $\boldsymbol{G} \succ \boldsymbol{O}$.*

*Proof.* If $\boldsymbol{G}$ is not positive definite, any $\boldsymbol{X}$ does not satisfy (20). By taking the contrapositive, if $\boldsymbol{X}$ is the solution of (20), $\boldsymbol{G}$ must be positive definite. Conversely, if $\boldsymbol{G}$ is positive definite, $\boldsymbol{G}^{\frac{1}{2}} \succ \boldsymbol{O}$ exists and the equation (20) becomes $\boldsymbol{X}\boldsymbol{G}^{\frac{1}{2}}\boldsymbol{G}^{\frac{1}{2}}\boldsymbol{X} = \boldsymbol{Q}^{\frac{1}{2}}\boldsymbol{Q}^{\frac{1}{2}}$. We thus have $\boldsymbol{X}\boldsymbol{G}^{\frac{1}{2}} = \boldsymbol{Q}^{\frac{1}{2}}$ and the equation has the solution $\boldsymbol{X} = \boldsymbol{Q}^{\frac{1}{2}}\boldsymbol{G}^{-\frac{1}{2}} \in \mathbb{S}_{++}^N$. $\qquad\square$

**Proposition 4.3.** *Suppose $\boldsymbol{H}_M^{(t)} \succ \boldsymbol{O}$. Then, the CARE (19) has a solution in $\mathbb{S}_{++}^N$ if*

$$\mu^{(t)} > \max\{-1, -1/\lambda_{\max}(\boldsymbol{H}_M^{(t)}(\boldsymbol{L}^{(t)} + \boldsymbol{I}))\}, \tag{21}$$

*where $\lambda_{\max}(\boldsymbol{X})$ denotes the largest eigenvalue of $\boldsymbol{X}$.*

*Proof.* The right-hand side of the following CARE

$$\boldsymbol{L}\left\{\mu^{(t)}\boldsymbol{H}_M^{(t)} + (\boldsymbol{L}^{(t)} + \boldsymbol{I})^{-1}\right\}\boldsymbol{L} = (1 + \mu^{(t)})\boldsymbol{L}^{(t)}\boldsymbol{H}_M^{(t)}\boldsymbol{L}^{(t)} \tag{22}$$

is positive definite by the conditions. For $\mu^{(t)} \geq 0$, the solution of (22) immediately exists by Lemma 4.1. When $-1 < \mu^{(t)} < 0$, we can see that the solution exists if and only if

$$\mu^{(t)}\boldsymbol{H}_M^{(t)} + (\boldsymbol{L}^{(t)} + \boldsymbol{I})^{-1} \succ \boldsymbol{O}$$

also from Lemma 4.1. Then, we have

$$
\begin{aligned}
\mu^{(t)}\boldsymbol{H}_M^{(t)} + (\boldsymbol{L}^{(t)} + \boldsymbol{I})^{-1} \succ \boldsymbol{O} &\iff \boldsymbol{H}_M^{(t)\frac{1}{2}}(\boldsymbol{I} + \mu^{(t)-1}\boldsymbol{H}_M^{(t)-\frac{1}{2}}(\boldsymbol{L}^{(t)} + \boldsymbol{I})^{-1}\boldsymbol{H}_M^{(t)-\frac{1}{2}})\boldsymbol{H}_M^{(t)\frac{1}{2}} \prec \boldsymbol{O} \\
&\iff \boldsymbol{I} + \mu^{(t)-1}\boldsymbol{H}_M^{(t)-\frac{1}{2}}(\boldsymbol{L}^{(t)} + \boldsymbol{I})^{-1}\boldsymbol{H}_M^{(t)-\frac{1}{2}} \prec \boldsymbol{O} \\
&\iff \mu^{(t)}\boldsymbol{I} \succ -\boldsymbol{H}_M^{(t)-\frac{1}{2}}(\boldsymbol{L}^{(t)} + \boldsymbol{I})^{-1}\boldsymbol{H}_M^{(t)-\frac{1}{2}} \\
&\iff \mu^{(t)}\boldsymbol{I} \succ -\boldsymbol{H}_M^{(t)-1}(\boldsymbol{L}^{(t)} + \boldsymbol{I})^{-1} \\
&\iff \mu^{(t)} > \lambda_{\max}(-\boldsymbol{H}_M^{(t)-1}(\boldsymbol{L}^{(t)} + \boldsymbol{I})^{-1}) \\
&\iff \mu^{(t)} > -1/\lambda_{\max}(\boldsymbol{H}_M^{(t)}(\boldsymbol{L}^{(t)} + \boldsymbol{I})).
\end{aligned}
$$

$\qquad\square$

In the accelerated algorithm, the inequality (21) should be satisfied strictly. Algorithm 2 shows the entire procedure of our MM-based learning with acceleration on the basis of Proposition 4.3. In Algorithm 2, we introduce two hyperparameters; one is a tolerance $\delta > 0$ that guarantees the inequality (21) strictly and $T_{\mathrm{acc}} \in \{0, 1, \ldots, T\}$ denotes up to how many iterations the acceleration is applied. We can automatically adjust the step size coefficient $\mu^{(t)}$ at each iteration in the algorithm with fixed $\delta > 0$, while user-defined fixed step size coefficients are used in the existing fixed-point algorithm (Mariet & Sra, 2015). In the resulting algorithm, we decide the step size by

$$\mu = \min \left\{ \max \left\{ -1/\lambda_{\max} \left( \boldsymbol{H}_M^{(t)}(\boldsymbol{L}^{(t)} + \boldsymbol{I}) \right), -1 \right\} + \delta, 0 \right\}$$

to ensure (21) and prevent $\mu^{(t)} > 0$, which may provide monotonic but slower convergence than $\mu^{(t)} = 0$. Since only the largest eigenvalue of $\boldsymbol{H}_M^{(t)}(\boldsymbol{L}^{(t)} + \boldsymbol{I})$ incorporates in the inequality (21), determining $\mu^{(t)}$ takes less computational time than solving the CARE.

## 5 Experiments

### 5.1 Experimental Settings

We evaluate performance of the learning methods for full-rank DPPs through experiments on synthetic and real-world datasets. For references, we take the fixed-point algorithm (FP) (Mariet & Sra, 2015) and Adam (Kingma & Ba, 2015) as a representative gradient-based method. For Adam, we factorize the kernel matrix as $\boldsymbol{L} = \boldsymbol{V}\boldsymbol{V}^\top$ by $\boldsymbol{V} \in \mathbb{R}^{N \times N}$ and optimize $\boldsymbol{V}$ as with the low-rank DPPs (Gartrell et al., 2017; Osogami et al., 2018). We adopt full-batch learning for all the algorithms.

We provide the following two initialization schemes with reference to (Mariet & Sra, 2015):

- `WISHART`: We sample an initial value from the Wishart distribution as $\boldsymbol{L}^{(0)} \sim \mathcal{W}(N, \boldsymbol{I})/N$.

- `BASIC`: We uniformly sample $v_{ij}^{(0)} \sim \mathcal{U}(0, \sqrt{2}/N)$ for $i, j = 1, 2, \ldots, N$ and initialize as $\boldsymbol{L}^{(0)} = \boldsymbol{V}^{(0)}\boldsymbol{V}^{(0)\top}$.

The `WISHART` initialization provides a near-identity matrix, while `BASIC` provides a unstructured matrix for $\boldsymbol{L}^{(0)}$.

We adopt the acceleration schemes for each algorithm. We set the step size $a = 1.3$ for the fixed-point algorithm[2] and the tolerance $\delta = 0.15$ for the proposed MM algorithm. For $T_{\mathrm{acc}} < t$, we use the default parameter $a = 1$ for the fixed-point algorithm, which monotonically increases the objective but no acceleration is applied, and the same way is used for the proposed MM. In Adam optimization, we employ the default values $\beta_1 = 0.999, \beta_2 = 0.9$ for the decay rates, and the machine epsilon $\epsilon = 10^{-8}$. The acceleration steps $T_{\mathrm{acc}}$ of the fixed-point and MM algorithms and the learning rate $\eta$ of Adam are set to be different with the initialization schemes: $T_{\mathrm{acc}} = 5, \eta = 0.1$ for `WISHART` initialization and $T_{\mathrm{acc}} = 10, \eta = 0.01$ for `BASIC` initialization.

In each experiment, we stop the learning when the criterion $\frac{|f(\boldsymbol{L}^{(t)}) - f(\boldsymbol{L}^{(t-1)})|}{|f(\boldsymbol{L}^{(t-1)})|} \leq \delta_{\mathrm{tol}}$ is satisfied. We set $\delta_{\mathrm{tol}} = 10^{-4}$ as the relative tolerance for all the experiments reported below. We implemented all the experiments in Julia, and all our experiments were run on a Linux Mint system with 32GB of RAM and an Intel Core i9-10900K CPU @ 3.70GHz.

### 5.2 Datasets

We compare the learning algorithms with the following three datasets.

---

[2]This is a possibly large value that does not fail optimization in our datasets.

**Synthetic**

We make true parameters as $\boldsymbol{L}^* = \boldsymbol{V}^* \boldsymbol{V}^{*\top}$ with $v_{ij}^* \sim \mathcal{U}(0, 10/N)$ for $i, j = 1, 2, \ldots, N$, and sample $M$ realizations from the DPP $P_{\boldsymbol{L}^*}(\cdot)$. We consider three different problem sizes: $(N, M) = (32, 2{,}500)$, $(N, M) = (32, 10{,}000)$, and $(N, M) = (128, 2{,}500)$. Because the true parameters are constructed from the uniform distribution, they are likely to have no clear structure. Using this `Synthetic` dataset, we test the general applicability of our method.

In `Synthetic`, true parameters $\boldsymbol{L}^*$ are available; we assess goodness of estimation using not only log-likelihoods but also the von Neumann divergences $D_{\mathrm{vN}}(\boldsymbol{L}, \boldsymbol{L}^*) = \mathrm{tr}(\boldsymbol{L} \log \boldsymbol{L} - \boldsymbol{L} \log \boldsymbol{L}^* - \boldsymbol{L} + \boldsymbol{L}^*)$, which is a Bregman divergence for positive definite matrices.

**Nottingham**

We apply our method to the `Nottingham` music dataset[3], which was used in (Boulanger-Lewandowski et al., 2012; Osogami et al., 2018). The dataset contains more than 1,000 folk tracks in the ABC format in which a sequence of chords is stored. We treat each chord in the tracks as an i.i.d. sample of a DPP on the ground set $\{1, 2, \ldots, 88\}$, where $N = 88$ is the number of keys. We randomly pick 25 tracks and that yields $M = 6{,}364$ samples on average.

In `Nottingham`, there is large disparity in the probability of each item appearing, with very low- and high-pitched keys being rarely used. Moreover, music theory prohibits certain key combinations within a chord. From these facts, the optimal $\boldsymbol{L}^*$ of the `Nottingham` dataset is expected to have unknown but particular structure.

**Amazon Baby Registry**

`Amazon baby registry` has served as a benchmark for learning methods of DPPs since (Gillenwater et al., 2014). It contains 13 categories of child care products, including "feeding" and "carseats," and on average, has $N = 71$ items and $M = 8{,}585$ samples, respectively. We run our experiment on each of the 13 categories to assess performance of the learning methods for medium-sized recommender systems.

### 5.3 Experimental Results

**Synthetic**

The final mean log-likelihoods, runtimes, and von-Neumann divergence values of the `Synthetic` datasets with the acceleration are presented in Table 1. For each experiment, we conducted 30 trials with different $\boldsymbol{L}^*$ and $\boldsymbol{L}^{(0)}$ and calculated the average and standard deviation. As shown in Table 1, our method (MM) achieves the best runtimes for all the settings. While the final log-likelihood values are almost equivalent by the algorithms in `WISHART` initialization, those obtained by the proposed MM tend to be larger in `BASIC` initialization. Furthermore, our method also produces the best von Neumann divergences $D_{\mathrm{vN}}$ with `BASIC` initialization and moderately performs with `WISHART` initialization. The results show good stablity of our method; the proposed algorithm is considered to be favorable in standard situations. The result of the `Synthetic` datasets without the acceleration is also shown in Appendix C.

In Figure 2, we show the learning curves with and without acceleration. While the fixed-point algorithm convergences stably yet slightly slow without the acceleration, the accelerated version becomes competitive in `WISHART` initialization. The Adam optimizer may temporarily fall into poor local optima, depending on the initial value. On the other hand, the proposed MM algorithm consistently indicates stable and rapid convergence both with and without the acceleration.

---

[3]Available at `https://abc.sourceforge.net/NMD/`.

Table 1: Final mean log-likelihoods, runtimes, and von Neumann divergences $D_{\mathrm{vN}}(\boldsymbol{L}, \boldsymbol{L}^*)$ of the `Synthetic` datasets. Each value is computed from the average or standard deviation of 30 trials with the accelerated settings.

| Data Size | Method | WISHART | | | BASIC | | |
|---|---|---|---|---|---|---|---|
| | | Log-likelihood | Runtime (s) | vN div. | Log-likelihood | Runtime (s) | vN div. |
| $N = 32$ $M = 2{,}500$ | FP | $-15.58 \pm 0.15$ | $0.39 \pm 0.03$ | $\mathbf{38.22} \pm 1.85$ | $-15.61 \pm 0.20$ | $1.46 \pm 0.31$ | $41.39 \pm 4.17$ |
| | Adam | $\mathbf{-15.55} \pm 0.16$ | $0.36 \pm 0.23$ | $56.77 \pm 9.40$ | $-15.64 \pm 0.34$ | $0.71 \pm 0.37$ | $33.42 \pm 3.13$ |
| | MM | $-15.58 \pm 0.15$ | $\mathbf{0.18} \pm 0.07$ | $42.63 \pm 2.38$ | $\mathbf{-15.45} \pm 0.20$ | $\mathbf{0.21} \pm 0.03$ | $\mathbf{30.16} \pm 1.97$ |
| $N = 32$ $M = 10{,}000$ | FP | $\mathbf{-15.58} \pm 0.17$ | $1.32 \pm 0.14$ | $\mathbf{38.05} \pm 2.01$ | $-15.71 \pm 0.14$ | $5.59 \pm 0.85$ | $40.79 \pm 3.29$ |
| | Adam | $\mathbf{-15.58} \pm 0.17$ | $1.23 \pm 0.56$ | $49.35 \pm 4.54$ | $-15.70 \pm 0.22$ | $3.05 \pm 1.45$ | $32.50 \pm 2.11$ |
| | MM | $\mathbf{-15.58} \pm 0.18$ | $\mathbf{0.48} \pm 0.09$ | $42.53 \pm 2.43$ | $\mathbf{-15.55} \pm 0.14$ | $\mathbf{0.77} \pm 0.09$ | $\mathbf{29.75} \pm 1.52$ |
| $N = 128$ $M = 2{,}500$ | FP | $-30.14 \pm 0.18$ | $3.36 \pm 0.22$ | $\mathbf{36.17} \pm 0.45$ | $-30.34 \pm 0.19$ | $6.37 \pm 0.48$ | $52.62 \pm 1.74$ |
| | Adam | $-30.18 \pm 0.22$ | $2.50 \pm 0.40$ | $44.88 \pm 1.53$ | $-30.46 \pm 1.08$ | $2.30 \pm 0.56$ | $39.44 \pm 5.54$ |
| | MM | $\mathbf{-30.11} \pm 0.18$ | $\mathbf{0.69} \pm 0.05$ | $42.54 \pm 0.53$ | $\mathbf{-30.08} \pm 0.19$ | $\mathbf{1.27} \pm 0.21$ | $\mathbf{32.15} \pm 0.55$ |

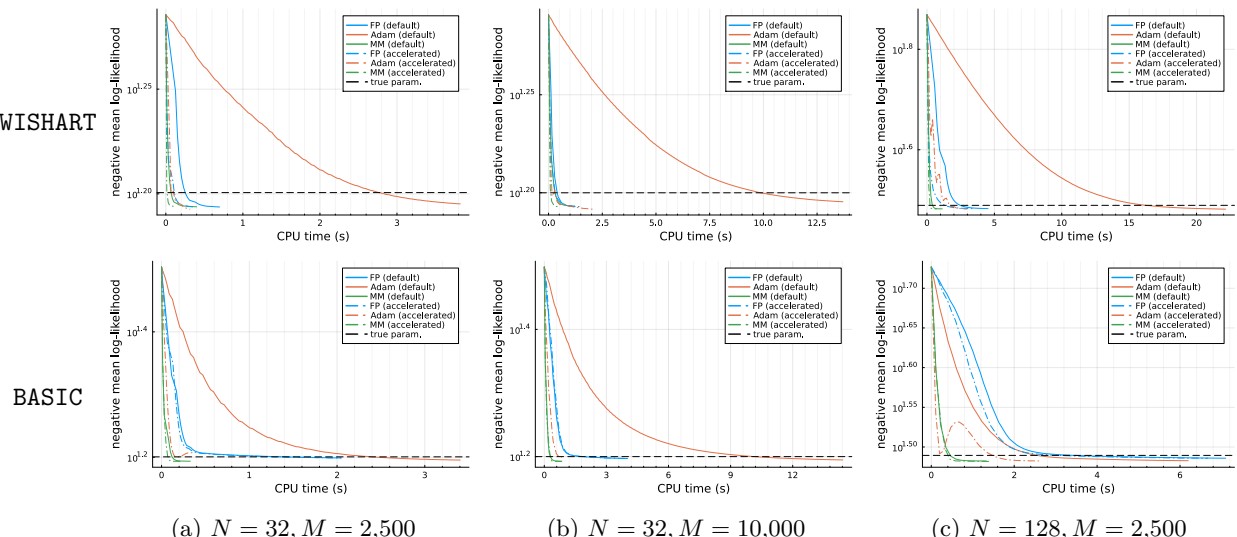

(a) $N = 32, M = 2{,}500$      (b) $N = 32, M = 10{,}000$      (c) $N = 128, M = 2{,}500$

Figure 2: Learning curves of the `Synthetic` datasets. Results with the default parameters ($T_{\mathrm{acc}} = 0$ for fixed-point and MM, and $\eta = 0.001$ for Adam) are also shown.

**Nottingham**

The results of the `Nottingham` dataset with the acceleration are presented in Table 2, and the learning curves with and without acceleration are showed in Figure 3. The convergence of Adam is remarkably rapid in the `Nottingham` dataset.

Under the `BASIC` initialization, the fixed-point and MM algorithms get stuck in poor local optima. Since the optimal $\boldsymbol{L}^*$ is considered to have a particular structure, the `BASIC` initialization may not be compatible with `Nottingham`. We can also find the acceleration scheme of the MM algorithm does not perform well in Figure 3 (see also the result without the acceleration shown in Appendix C). This may be because the assumption $\bigcup_{m=1}^{M} \mathcal{A}_m = \mathcal{Y}$ for the accelerated MM is not satisfied in the `Nottingham` dataset.

**Amazon Baby Registry**

In Table 3, we show the results with the accelerated algorithms in all the 13 categories of `Amazon baby registry`. Overall, our algorithm achieves moderately better log-likelihood values and outstanding convergence speeds in most categories. Adam tends to produce the best final log-likelihoods but they are not statistically significant in most cases. Especially, when the sample size is relatively large, such as $M > 10{,}000$,

Table 2: Final mean log-likelihoods and runtimes of the `Nottigham` dataset. Each value is computed from the average or standard deviation of 30 trials with the accelerated settings.

| Method | WISHART | | BASIC | |
|---|---|---|---|---|
| | Log-likelihood | Runtime (s) | Log-likelihood | Runtime (s) |
| FP | $-8.31 \pm 0.22$ | $40.68 \pm 2.86$ | $-10.13 \pm 0.27$ | $33.19 \pm 8.08$ |
| Adam | $\mathbf{-7.84} \pm 1.02$ | $\mathbf{9.01} \pm 1.59$ | $\mathbf{-7.92} \pm 0.64$ | $\mathbf{21.05} \pm 6.75$ |
| MM | $-9.51 \pm 0.24$ | $19.69 \pm 4.89$ | $-9.58 \pm 0.21$ | $19.11 \pm 5.49$ |

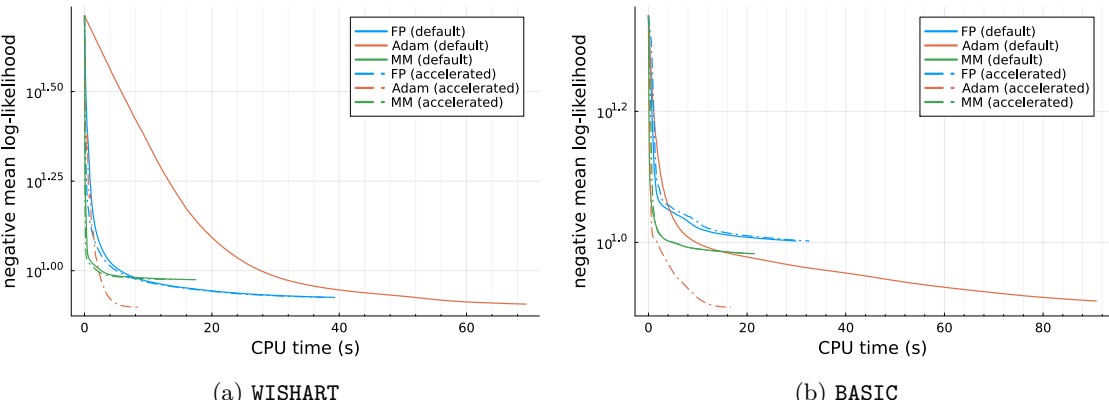

(a) WISHART       (b) BASIC

Figure 3: Learning curves of the `Nottingham` dataset. Results with the default parameters ($T_{\mathrm{acc}} = 0$ for fixed-point and MM, and $\eta = 0.001$ for Adam) are also shown.

our algorithm outperforms in the convergence speed that is about 5-10 times faster than the fixed-point algorithm.

Although the convergences of the MM algorithm seems to be slow in some of the smaller categories in Table 3, that is not very serious. In these cases, the MM algorithm quickly reaches a near optimum value, but takes longer to meet the stopping criterion. By managing the stopping criterion, we may be able to stop its learning much earlier.

# 6 Conclusion and Future Work

In this paper, we developed an efficient learning method for full-rank DPPs based on the MM algorithm. Compared with the existing methods, our algorithm has many advantages: it has guaranteed convergence and monotonicity, requires no bothersome hyperparameters, convergences rapidly and stably, and is easy to implement. Upon considering the performance of our algorithm, we revealed that our algorithm provides a locally tighter minorizer than the existing method. We also assessed the empirical performance of our method through experiments on both synthetic and real-world datasets, outperforming in terms of convergence speed and reaching a better estimate in most experimental settings.

We believe that our algorithm is a strong candidate for learning full-rank DPPs at the present moment, but there is still much future work to be done. First, we need to deepen our understanding of the performance of our method. Proposition 3.3 partially addresses this question, but it is just an implication. One considerable future direction is establishing general recipes for comparing MM with different minorizers. Second, scaling up our method for large $N$ is a crucial issue. Several numerical algorithms for solving large-sized CARE (12) have been proposed based on some structure of a problem: low-rank structure and/or sparsity (Bini et al., 2011; Simoncini, 2016). On the other hand, our CARE (9) formed by full-rank and dense matrices, therefore, exploring a good CARE solver is considered to be an essential task.

Table 3: Final mean log-likelihoods and runtimes of the `Amazon baby registry` dataset. Each value is computed from the average or standard deviation of 30 trials with the accelerated settings and initialized by `WISHART`.

| Category | Method | Log-likelihood | Runtime (s) | Category | Method | Log-likelihood | Runtime (s) |
|---|---|---|---|---|---|---|---|
| Apparel $N = 100$ $M = 14{,}970$ | FP | $-10.20 \pm 0.00$ | $24.54 \pm 0.69$ | Gear $N = 100$ $M = 16{,}823$ | FP | $-9.27 \pm 0.00$ | $30.54 \pm 0.90$ |
| | Adam | $\mathbf{-10.08} \pm 0.26$ | $17.99 \pm 2.61$ | | Adam | $\mathbf{-9.16} \pm 0.41$ | $25.85 \pm 5.74$ |
| | MM | $-10.17 \pm 0.00$ | $\mathbf{3.13} \pm 0.30$ | | MM | $-9.24 \pm 0.00$ | $\mathbf{2.02} \pm 0.38$ |
| Bath $N = 100$ $M = 14{,}542$ | FP | $-8.79 \pm 0.00$ | $26.51 \pm 0.47$ | Health $N = 62$ $M = 14{,}057$ | FP | $-7.59 \pm 0.00$ | $13.22 \pm 0.35$ |
| | Adam | $\mathbf{-8.72} \pm 0.79$ | $17.97 \pm 4.84$ | | Adam | $\mathbf{-7.37} \pm 0.27$ | $10.06 \pm 1.66$ |
| | MM | $-8.75 \pm 0.00$ | $\mathbf{2.06} \pm 0.47$ | | MM | $-7.55 \pm 0.00$ | $\mathbf{2.16} \pm 0.44$ |
| Bedding $N = 100$ $M = 16{,}370$ | FP | $-8.79 \pm 0.00$ | $32.23 \pm 0.73$ | Media $N = 58$ $M = 5{,}904$ | FP | $-8.56 \pm 0.00$ | $4.01 \pm 0.67$ |
| | Adam | $\mathbf{-8.59} \pm 0.18$ | $23.26 \pm 1.31$ | | Adam | $\mathbf{-8.39} \pm 0.16$ | $2.97 \pm 1.07$ |
| | MM | $-8.77 \pm 0.00$ | $\mathbf{4.79} \pm 1.10$ | | MM | $-8.52 \pm 0.01$ | $\mathbf{1.75} \pm 0.75$ |
| Carseats $N = 34$ $M = 7{,}566$ | FP | $-5.18 \pm 0.06$ | $5.04 \pm 3.27$ | Safety $N = 36$ $M = 8{,}892$ | FP | $-4.76 \pm 0.16$ | $8.93 \pm 7.46$ |
| | Adam | $\mathbf{-4.82} \pm 0.29$ | $\mathbf{2.03} \pm 0.33$ | | Adam | $\mathbf{-4.30} \pm 0.00$ | $\mathbf{2.28} \pm 0.10$ |
| | MM | $-5.00 \pm 0.05$ | $4.96 \pm 1.45$ | | MM | $-4.57 \pm 0.05$ | $6.19 \pm 2.10$ |
| Diaper $N = 100$ $M = 16{,}759$ | FP | $-10.71 \pm 0.00$ | $27.16 \pm 0.83$ | Strollers $N = 40$ $M = 7{,}393$ | FP | $-5.66 \pm 0.06$ | $4.58 \pm 3.21$ |
| | Adam | $\mathbf{-10.61} \pm 0.35$ | $25.75 \pm 5.96$ | | Adam | $\mathbf{-5.25} \pm 0.38$ | $\mathbf{2.35} \pm 0.39$ |
| | MM | $-10.67 \pm 0.00$ | $\mathbf{3.21} \pm 0.53$ | | MM | $-5.46 \pm 0.05$ | $6.12 \pm 2.39$ |
| Feeding $N = 100$ $M = 19{,}001$ | FP | $-12.17 \pm 0.00$ | $28.97 \pm 0.36$ | Toys $N = 62$ $M = 10{,}073$ | FP | $-8.10 \pm 0.00$ | $7.65 \pm 0.71$ |
| | Adam | $-12.17 \pm 0.27$ | $18.38 \pm 5.11$ | | Adam | $\mathbf{-7.94} \pm 0.27$ | $5.77 \pm 1.25$ |
| | MM | $\mathbf{-12.15} \pm 0.00$ | $\mathbf{3.11} \pm 0.39$ | | MM | $-8.07 \pm 0.00$ | $\mathbf{1.45} \pm 0.34$ |
| Furniture $N = 32$ $M = 7{,}093$ | FP | $-4.86 \pm 0.13$ | $4.93 \pm 4.75$ | | | | |
| | Adam | $\mathbf{-4.40} \pm 0.00$ | $\mathbf{1.88} \pm 0.05$ | | | | |
| | MM | $-4.65 \pm 0.05$ | $5.37 \pm 1.78$ | | | | |

**Acknowledgements**

We thank anonymous reviewers for insightful comments and suggestions. Part of this work is supported by NEDO grant number JPNP18002, JST grant number JPMJFS2136, JST CREST JPMJCR2015, and JSPS KAKENHI JP22H03653.

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

## A    Proof of Proposition 3.3

### A.1    Derivation of Equation (15)

In (14), the constant term is given by

$$\xi(\boldsymbol{L}^{(t)}) = \frac{1}{M} \sum_{m=1}^{M} \left\{ \log \det(\boldsymbol{U}_{\mathcal{A}_m} \boldsymbol{L}^{(t)} \boldsymbol{U}_{\mathcal{A}_m}^{\top}) + |\mathcal{A}_m| \right\} + \log \det\{(\boldsymbol{L}^{(t)} + \boldsymbol{I})^{-1} \boldsymbol{L}^{(t)}\} + \mathrm{tr}\{(\boldsymbol{L}^{(t)} + \boldsymbol{I})^{-1}\}.$$

By the following inequality from the Taylor expansion

$$- \log \det(\boldsymbol{L}^{(t)}) \geq - \log \det(\boldsymbol{L}) - \mathrm{tr}\{(\boldsymbol{L}^{-1}(\boldsymbol{L}^{(t)} - \boldsymbol{L})\},$$

we have

$$
\begin{aligned}
g(\boldsymbol{L}|\boldsymbol{L}^{(t)}) - h(\boldsymbol{L}|\boldsymbol{L}^{(t)}) &= \operatorname{tr}\{(\boldsymbol{L}^{(t)} + \boldsymbol{I})^{-1}(\boldsymbol{L}^{-1}\boldsymbol{L}^{(t)} - \boldsymbol{I} - \boldsymbol{L} + \boldsymbol{L}^{(t)})\} + \log\det(\boldsymbol{L}) - \log\det(\boldsymbol{L}^{(t)}) \\
&\geq \operatorname{tr}\{(\boldsymbol{L}^{(t)} + \boldsymbol{I})^{-1}(\boldsymbol{L}^{-1}\boldsymbol{L}^{(t)} - \boldsymbol{I} - \boldsymbol{L} + \boldsymbol{L}^{(t)})\} - \operatorname{tr}\{\boldsymbol{L}^{-1}(\boldsymbol{L}^{(t)} - \boldsymbol{L})\} \\
&= \operatorname{tr}\{(\boldsymbol{L}^{(t)} + \boldsymbol{I})^{-1}(\boldsymbol{L}^{-1}\boldsymbol{L}^{(t)} - \boldsymbol{L} + 2\boldsymbol{L}^{(t)})\} - \operatorname{tr}\{\boldsymbol{L}^{-1}\boldsymbol{L}^{(t)}\} - N + N \\
&= \operatorname{tr}\{(\boldsymbol{L}^{(t)} + \boldsymbol{I})^{-1}(2\boldsymbol{L}^{(t)} - \boldsymbol{L} - \boldsymbol{L}^{(t)}\boldsymbol{L}^{-1}\boldsymbol{L}^{(t)})\}.
\end{aligned}
$$

### A.2 Derivation of Equation (16)

Let $\boldsymbol{L} = \boldsymbol{L}^{(t)} + \delta\boldsymbol{M}$, where $\delta > 0$ is a sufficient small coefficient and $\boldsymbol{M}$ is a symmetric matrix whose all eigenvalues are in $[-1, 1]$. Then, we can approximate the matrix inverse as $\boldsymbol{L}^{-1} = (\boldsymbol{L}^{(t)} + \delta\boldsymbol{M})^{-1} \approx \boldsymbol{L}^{(t)-1} - \delta\boldsymbol{L}^{(t)-1}\boldsymbol{M}\boldsymbol{L}^{(t)-1}$ by the Taylor expansion. Using this approximation, we have

$$
\begin{aligned}
2\boldsymbol{L}^{(t)} - \boldsymbol{L} - \boldsymbol{L}^{(t)}\boldsymbol{L}^{-1}\boldsymbol{L}^{(t)} &= 2\boldsymbol{L}^{(t)} - (\boldsymbol{L}^{(t)} + \delta\boldsymbol{M}) - \boldsymbol{L}^{(t)}(\boldsymbol{L}^{(t)} + \delta\boldsymbol{M})^{-1}\boldsymbol{L}^{(t)} \\
&\approx 2\boldsymbol{L}^{(t)} - (\boldsymbol{L}^{(t)} + \delta\boldsymbol{M}) - \boldsymbol{L}^{(t)}(\boldsymbol{L}^{(t)-1} - \delta\boldsymbol{L}^{(t)-1}\boldsymbol{M}\boldsymbol{L}^{(t)-1})\boldsymbol{L}^{(t)} \\
&= \boldsymbol{O}.
\end{aligned}
$$

## B Proof of Proposition 4.1

*Proof.* $f_{\mu^{(t)}}(\boldsymbol{L}|\boldsymbol{L}^{(t)})$ can be minorized as:

$$f_{\mu^{(t)}}(\boldsymbol{L}|\boldsymbol{L}^{(t)})$$

$$
= \frac{1}{M}\sum_{m=1}^{M}(\log\det([\boldsymbol{L}]_{\mathcal{A}_m}) - \underbrace{\mu^{(t)}\log\det([\boldsymbol{L}]_{\mathcal{A}_m})}_{\substack{\text{majorizing by (3) w/}\\ \boldsymbol{X}\to[\boldsymbol{L}]_{\mathcal{A}_m},\\ \boldsymbol{Y}\to[\boldsymbol{L}^{(t)}]_{\mathcal{A}_m}}} - \mu^{(t)}\operatorname{tr}([\boldsymbol{L}]_{\mathcal{A}_m}^{-1}[\boldsymbol{L}^{(t)}]_{\mathcal{A}_m})) - \underbrace{\log\det(\boldsymbol{L} + \boldsymbol{I})}_{\substack{\text{majorizing by (3) w/}\\ \boldsymbol{X}\to\boldsymbol{L}+\boldsymbol{I},\\ \boldsymbol{Y}\to\boldsymbol{L}^{(t)}+\boldsymbol{I}}} + \text{const.}
$$

$$
\geq \frac{1}{M}\sum_{m=1}^{M}(\underbrace{\log\det([\boldsymbol{L}]_{\mathcal{A}_m})}_{\substack{\text{minorizing by (4) w/}\\ \boldsymbol{X}\to[\boldsymbol{L}]_{\mathcal{A}_m},\\ \boldsymbol{Y}\to[\boldsymbol{L}^{(t)}]_{\mathcal{A}_m}}} - \mu^{(t)}\operatorname{tr}([\boldsymbol{L}^{(t)}]_{\mathcal{A}_m}^{-1}[\boldsymbol{L}]_{\mathcal{A}_m}) - \mu^{(t)}\operatorname{tr}([\boldsymbol{L}]_{\mathcal{A}_m}^{-1}[\boldsymbol{L}^{(t)}]_{\mathcal{A}_m})) - \operatorname{tr}\{(\boldsymbol{L} + \boldsymbol{I})^{-1}\boldsymbol{L}\} + \text{const.}
$$

$$
\geq -\frac{1}{M}\sum_{m=1}^{M}((1 + \mu^{(t)})\underbrace{\operatorname{tr}([\boldsymbol{L}]_{\mathcal{A}_m}^{-1}[\boldsymbol{L}^{(t)}]_{\mathcal{A}_m})}_{\substack{\text{majorizing by (7) w/}\\ \boldsymbol{A}\boldsymbol{P}\boldsymbol{A}^\top\to[\boldsymbol{L}]_{\mathcal{A}_m},\\ \boldsymbol{S}\to[\boldsymbol{L}^{(t)}]_{\mathcal{A}_m}}} + \mu^{(t)}\operatorname{tr}([\boldsymbol{L}^{(t)}]_{\mathcal{A}_m}^{-1}[\boldsymbol{L}]_{\mathcal{A}_m})) - \operatorname{tr}\{(\boldsymbol{L} + \boldsymbol{I})^{-1}\boldsymbol{L}\} + \text{const.}
$$

$$
\geq -\frac{1 + \mu^{(t)}}{M}\sum_{m=1}^{M}\operatorname{tr}(\boldsymbol{L}^{(t)}\boldsymbol{U}_{\mathcal{A}_m}^\top[\boldsymbol{L}^{(t)}]_{\mathcal{A}_m}^{-1}\boldsymbol{U}_{\mathcal{A}_m}\boldsymbol{L}^{(t)}\boldsymbol{L}^{-1})
$$

$$
- \frac{\mu^{(t)}}{M}\sum_{m=1}^{M}\operatorname{tr}([\boldsymbol{L}^{(t)}]_{\mathcal{A}_m}^{-1}[\boldsymbol{L}]_{\mathcal{A}_m}) - \operatorname{tr}\{(\boldsymbol{L}^{(t)} + \boldsymbol{I})^{-1}\boldsymbol{L}\} + \text{const.}
$$

$$
= g_{\mu^{(t)}}(\boldsymbol{L}|\boldsymbol{L}^{(t)}).
$$

$\square$

## C Additional Experimental Results

Table C.1 shows the learning result of the `Synthetic` dataset with the default (non-accelerated) settings. $T_{\text{acc}} = 0$ for fixed-point and MM and $\eta = 0.001$ for Adam are used as the default settings. We find the

Table C.1: Final mean log-likelihoods, runtimes, and von Neumann divergences $D_{\mathrm{vN}}(\boldsymbol{L}, \boldsymbol{L}^*)$ of the `Synthetic` datasets. Each value is computed from the average or standard deviation of 30 trials with the non-accelerated settings.

| Data Size | Method | WISHART | | | BASIC | | |
|---|---|---|---|---|---|---|---|
| | | Log-likelihood | Runtime (s) | vN div. | Log-likelihood | Runtime (s) | vN div. |
| $N = 32$ $M = 2{,}500$ | FP | $-15.58 \pm 0.15$ | $0.43 \pm 0.06$ | $38.20 \pm 1.84$ | $-15.61 \pm 0.21$ | $1.69 \pm 0.22$ | $42.19 \pm 4.19$ |
| | Adam | $-15.63 \pm 0.15$ | $3.29 \pm 0.25$ | $63.07 \pm 5.71$ | $-15.54 \pm 0.20$ | $3.41 \pm 0.17$ | $29.51 \pm 1.85$ |
| | MM | $-15.58 \pm 0.15$ | $0.40 \pm 0.06$ | $43.94 \pm 2.62$ | $-15.46 \pm 0.20$ | $0.32 \pm 0.03$ | $30.15 \pm 1.99$ |
| $N = 32$ $M = 10{,}000$ | FP | $-15.58 \pm 0.18$ | $1.32 \pm 0.17$ | $38.03 \pm 2.00$ | $-15.72 \pm 0.14$ | $5.53 \pm 0.96$ | $41.61 \pm 3.30$ |
| | Adam | $-15.66 \pm 0.17$ | $12.83 \pm 1.11$ | $61.41 \pm 5.40$ | $-15.63 \pm 0.14$ | $14.59 \pm 0.56$ | $29.08 \pm 1.23$ |
| | MM | $-15.58 \pm 0.18$ | $1.22 \pm 0.16$ | $43.95 \pm 2.58$ | $-15.56 \pm 0.14$ | $1.00 \pm 0.11$ | $29.74 \pm 1.47$ |
| $N = 128$ $M = 2{,}500$ | FP | $-30.14 \pm 0.18$ | $3.70 \pm 0.22$ | $36.20 \pm 0.44$ | $-30.35 \pm 0.19$ | $6.56 \pm 0.45$ | $53.47 \pm 1.79$ |
| | Adam | $-30.05 \pm 0.19$ | $24.05 \pm 0.78$ | $85.47 \pm 4.02$ | $-30.12 \pm 0.19$ | $5.84 \pm 0.23$ | $33.44 \pm 0.56$ |
| | MM | $-30.11 \pm 0.18$ | $1.39 \pm 0.08$ | $44.68 \pm 0.62$ | $-30.10 \pm 0.19$ | $1.29 \pm 0.07$ | $32.26 \pm 0.50$ |

Table C.2: Final mean log-likelihoods and runtimes of the `Nottigham` dataset. Each value is computed from the average or standard deviation of 30 trials with the non-accelerated settings.

| Method | WISHART | | BASIC | |
|---|---|---|---|---|
| | Log-likelihood | Runtime (s) | Log-likelihood | Runtime (s) |
| FP | $-8.30 \pm 0.22$ | $40.92 \pm 3.01$ | $-10.14 \pm 0.28$ | $33.75 \pm 6.84$ |
| Adam | $-7.81 \pm 0.25$ | $68.12 \pm 4.94$ | $-8.02 \pm 0.26$ | $103.27 \pm 15.57$ |
| MM | $-9.51 \pm 0.25$ | $21.73 \pm 6.04$ | $-9.59 \pm 0.22$ | $19.95 \pm 3.59$ |

proposed MM algorithm with the default setting still performs better than the other algorithms with the accelerated settings, shown in Table 1.

Table C.2 shows the result of the `Nottingham` dataset with the default settings which is the same to the `Synthetic` experiments. In contrast to `Synthetic`, the performance of the MM algorithm with and without the acceleration is not much different (cf. Table 2). This may be due to the absence of the assumption required in the accelerated MM algorithm. We need that $\bigcup_m \mathcal{A}_m = \mathcal{Y}$ in Section 4, but `Nottingham` does not satisfy that as described in Section 5.

