# OpenReview forum: "Minorization-Maximization for Learning Determinantal Point Processes"
_TMLR — Accepted by TMLR_

### Review · Reviewer_ENwe · 2023-06-08

**Summary Of Contributions:**

This paper proposes a novel algorithm to learn the (full-rank) kernel matrix that parameterizes a determinantal point process (DPP).

Taking advantage of the fact that the DPP log-likelihood is a sum of concave and convex functions, the authors propose a learning procedure based on minorization-maximization (MM). The authors compare their method to (Mariet & Sra, 2015) -which relied upon a similar decomposition of the log-likelihood - and show that the MM procedure relies upon a tighter approximation to the true log-likelihood in the neighborhood of the current iteration.

Each MM learning iteration has a computational cost equal to that of (Mariet & Sra), but the authors show on synthetic and the Amazon Baby registry (a canonical experimental dataset for DPPs) datasets that MM converges in significantly less CPU time.

**Audience:**

Yes

**Claims And Evidence:**

Yes

**Requested Changes:**

Please address the comments above.

**Strengths And Weaknesses:**

# Strengths
- The proposed algorithm is elegant, and also subsumes prior work as a special case
- The algorithm requires no hyperparameters to tune, and is guaranteed to improve the log-likelihood at each step
- Under the evaluated experimental conditions, MM almost always significantly cuts down on the runtime of (Mariet & Sra, 2016).
- The experimental validation is well presented, and the conclusions are clear.

# Weaknesses
- Clarity: there are a couple of statements that should be clarified.
  + I believe that the concept of "negative dependence" (second paragraph of the introduction) is in fact a precise concept that is not equivalent to $P(A \cup B) < P(A) P(B)$ (c.f. _Negative dependence and the geometry of polynomials_, Borcea et al., 2007). Given the importance of negative dependence as a motivator for the use of DPPs in the community, I would recommend the authors use more precise terminology here.
  + Could you clarify the inequality $P(A \cup B) < P(A) P(B)$? It almost looks like the "negative association" condition (_Negative dependence and the geometry of polynomials_, Borcea et al., 2007), but that one requires a disjoint set of coordinates. Also, should it be a loose $\leq$, in the case where A and B are independent?
  + It might be worth clarifying for non-expert readers that the requirement $0 \leq K \leq I$ is necessary to ensure that the resulting probabilities $0 \leq P(K_S) \leq 1$.
  + Not all DPPs admit a formulation with kernel $L$; if I remember correctly, this is only true if $K$ has all eigenvalues < 1, which is equivalent to $P(\emptyset) > 0$.
  + At the end of the introduction, could you clarify why the concavity of the minorizer is important?

- Experimental validation
  + I am confused by the process by which the "best" performance is bolded in the tables. Often, different algorithms will have the exact same performance, but only one will be bolded (seemingly randomly, e.g., in the synthetic datasets).
  + Nottingham analysis: it seems like MM (and FP) converge poorly for both initializations, not just BASIC. I'd be interested in a more in-depth analysis of what could be causing this. The authors mention expecting a specific structure for the optimal kernel L. Could it also be that the MM-based algorithms (including FP) perform less well for large $N$, or that they converge to almost singular matrices?

# Questions for the authors
- Would it be possible to introduce a step size $a$ in the same way that (Mariet & Sra) do?
- Would MM be amenable to specific structures such as low-rank or Kronecker constraints?

---

> ### Author Response · Authors · 2023-09-24
> **Response to Reviewer ENwe**
>
> Thank you for the insightful comments. The following changes were made to the paper in order to fulfill your suggestions.
>
> ---
> ### Response to "Weaknesses"
> > I believe that the concept of "negative dependence" (second paragraph of the introduction) is...
>
> > Could you clarify the inequality $ P(A \cup B) < P(A) P(B)$? ...
>
>  [page 1, paragraph 2] We had a slight misunderstanding about the terminology of "negative dependence." Thank you for pointing out. We add the citations to your suggested paper and the Mariet's Ph.D. thesis (Mariet, 2019). To describe the negative dependence of DPPs, we use an example on pairwise negative correlation.
>
> > It might be worth clarifying for non-expert readers that the requirement $0 < K < I$ is necessary...
>
> [page 3, paragraph 3] We clarified the requirement.
>
> > Not all DPPs admit a formulation with kernel $L$;...
>
> [page 3, paragraph 3] We made it explicit that $I - K$ must be invertible to obtain $L$ from $K$.
>
> > At the end of the introduction, could you clarify why the concavity of the minorizer is important?
>
> [page 2, paragraph 4] We added the statement about the importance.
>
> > I am confused by the process by which the "best" performance is bolded in the tables....
>
> [Tables] This is due to differences below the significant digits. Because that was confusing as you say and just a negligible error, we took your comment in the new experiments.
>
> > Nottingham analysis: it seems like MM (and FP) converge poorly for both initializations, not just BASIC....
>
> We guess that other singular matrices also make convergence of MM poor (both with and without acceleration) because the feasible region of MM-based algorithms is $ \mathbb{S}^N_{++}$ and they assume $\cup^M_{m = 1} \mathcal{A}_m = \mathcal{Y}$. For large $N$, the issue could be likely to occur, but the complexity $\mathcal{O}(N^3)$ may arises before that. Relating this, the acceleration does not work in such a degenerated observations too. We added a comment about the acceleration in the updated paper.
>
>
> ---
> ### Response to "Questions for the authors"
> > Would it be possible to introduce a step size in the same way that (Mariet & Sra) do?
>
> [Section 4] Yes. Motivating this question, we have developed generalization of our MM algorithm to introducing a step size. Thank you for the nice comment!
>
> > Would MM be amenable to specific structures such as low-rank or Kronecker constraints?
>
> For low-rank parameterization, an MM-based optimization can also be applied but we think a straightforward extension of our strategy for low-rank kernels is difficult because it explicitly requires positive definiteness of $L$. For Kronecker-factorized DPPs, perhaps our strategy is applicable. Developing an alternating procedure for learning $A, B$ s.t. $L = A \otimes B$ based on MM is might be worth as future work.

---

### Review · Reviewer_qfjS · 2023-07-05

**Summary Of Contributions:**

The paper introduces a new method for learning full-rank Determinantal Point Processes (DPPs).

1. The paper's main contribution is introducing a simple learning recipe based on the minorization-maximization (MM) algorithm. The proposed method allows for efficient and stable learning, increasing the log-likelihood monotonically at each iteration.

1. The authors compare the (local) tightness of the minorizers between the existing and proposed methods. The existing fixed-point algorithm for DPPs (Mariet & Sra, 2015) can also be viewed as an MM algorithm. The results indicate that the proposed minorizer provides a tighter lower bound **locally** than the existing method. Furthermore, the proposed method's minorizer is concave; in contrast, the existing method maximizes a non-concave minorizer during each iteration.

1. The paper conducts experiments to evaluate learning algorithms for full-rank DPPs using synthetic and real-world datasets. The authors focus on convergence speed and performance in terms of log-likelihood to compare against existing methods.



**Audience:**

Yes

**Broader Impact Concerns:**

No concerns here.

**Claims And Evidence:**

No

**Requested Changes:**

Expanding on the weaknesses mentioned above:

* I believe using only $a=1$ for the fixed-point method is unfair. Looking at the paper by Mariet & Sra (2015), I noticed that they rarely set $a=1$. While their convergence guarantee is shown for $a=1$, they also suggest that values $a>1$ also often work well. I would help to see what happens with their method for, say, $a \in \\{1.3, 1.5, 2\\}$? Similarly, for the Adam method, it would help to see what happens for $\eta = 10^{-8}$. The authors state that they observe failures in the optimization for $\eta \geq 10^{-7}$, yet they decided to set $\eta = 10^{-9}$. Given that the proposed method barely improves over Adam in terms of runtime, I am curious if this change would result in Adam being faster than MM in more settings.

* While the code was easy to run, it would help to clarify a few things: I noticed that in the file `dpp_utils.jl` the function `mle_mm` does not call the function `compute_minorizer_mm`, which was confusing to me. Similarly, for the function `compute_minorizer_fp`. I am not an expert on Julia so I might have missed something here.

**Strengths And Weaknesses:**

### Strengths

* The learning problem under study is important. DPPs are relevant models in machine learning due to their ability to model negative dependence, thus, encouraging the occurrence of diverse subsets and making them suitable for various applications.

* The proposed approach results from a simple application of the MM algorithm. Thus, the paper is easy to follow.

* Practical applicability. The proposed learning algorithm offers several positive properties: easy to implement, hyperparameter-free, and well-behaved. It can be readily used for exact inference on small to medium-sized datasets, making it suitable for various machine learning tasks that require diverse subset generation.

* The code was easy to run and reproduce the experiments.

### Weaknesses

* The technical contribution, in general, is rather incremental.

* It is unclear how the proposed minorizer truly affects the convergence rate. The authors claim that their method attains faster convergence empirically. However, in several cases, such as in Table 3, I noticed that Adam obtains faster runtimes than MM (the proposed approach) and comparable log-likelihoods to MM.

* I believe the paper would benefit from additional proof and experiment details. For example, in the proof of Proposition 3.1, a matrix inequality is recalled from Sun et al. (2016) and Sun et al. (2017). It would be much easier to either provide an explicit reference to the inequality in those papers or elaborate more on what $A$ and $P_t$ are. For the latter, are A and $P_t$ any matrix, or are they both required to be positive definite? In the experiments, it was not clear whether batch or minibatch was used for Adam, and I had to look at the code to clear my doubts.

---

> ### Author Response · Authors · 2023-09-24
> **Response to Reviewer qfjS**
>
> Thank you for the suggestions and checking the script. We made revisions with respect to your comments.
>
> ---
> ### Response to "Weaknesses"
>
> > It is unclear how the proposed minorizer truly affects the convergence rate....
>
> [Tables] We have developed an accelerated version of the MM and we believe that it enhances the superiority. In the many well-suited datasets (Synthetic and Amazon), while the differences of log-likelihoods among the methods is not significant, our algorithm performs the best in convergence speed (with statistical significance).
>
> > I believe the paper would benefit from additional proof and experiment details.
>
> [page 4, Prop. 3.1; page 5, Prop. 3.2; page 10, paragraph 2] We added more details in the proofs of Prop. 3.1 and Prop. 3.2. In the section 5.1 (Experimental Settings), we clarified that full-batch learning is used.
>
> > I believe using only $a = 1$ for the fixed-point method is unfair....
>
> [Section 5] This concern may have been resolved by the updated experiments comparing the accelerated algorithms, in which $a = 1.3$ for the fixed-point algorithm and $\eta = 0.1$ or $0.01$ for Adam are used. The hyperparameters have been chosen as possibly large values which do not cause a convergence failure. We note that the reason that small learning rates like $\eta = 10^{-7}$ did not work is our implementation bug in Adam. In the updated experiments, $\eta$ within the standard range were used.
>
> > While the code was easy to run, it would help to clarify a few things:...
>
> Sorry for the confusing script. The functions `compute_minorizer_mm` and `compute_minorizer_fp` are used just for drawing Figure 1. Because the algorithms do not require to compute the values of their minorizers within the iterations, the functions are not called in the main routines.

---

### Review · Reviewer_EE9h · 2023-09-10

**Summary Of Contributions:**

This paper proposes a Minorization-Maximization (MM) style algorithm for learning a kernel matrix that represents a discrete determinantal point process (DPP) on $n$ points. That is, we are given a base set of size $n$, and a training dataset which is a list of subsets of $[n]$. For some kernel matrix $K$, we can use DPP sampling to sample some subset of $[n]$. Our goal is to learn a kernel matrix $K$ that, when using DDP sampling, maximizes the likelihood of the training dataset.

The proposed algorithm is an iterative nonconvex optimization algorithm, which requires inverting a matrix and solving a "continuous algebraic Ricatti equation" in each iteration. These linear algebra tasks are computable at the moderately-large-data scale, but not the truly big data scale, which the paper owns.

The paper has some theoretical and a good deal of experimental evidence. The theoretical results show that the approximation to the likelihood used by the proposed algorithm is more accurate than the lower bound used in a prior work's algorithm, but this claim only holds in a neighborhood of the current iterate. The experimental evidence suggests that the algorithm works well in practice, but the claim that it outperforms other algorithms it a bit messier.

**Audience:**

Yes

**Claims And Evidence:**

No

**Requested Changes:**

I listed my big picture qualms above, in the "strengths and weaknesses" section. I'll be extremely concrete here. Afterwards, I'll write the list of typos / minor recommended edits.

1. [Proof of Prop 3.1] Explain the inequalities used more. What is $P_t$, $R_t$, and $P$? What is the exact taylor approximation being used? What is happening in this proof? It's completely unfollowable.
1. [Proof of Prop 3.2] Consider showing more work in how you prove equation (8). It's not super important here, but this one is a bit more work to do on your own. Feel free to ignore this point.
1. [Proof of Corol 3.1] I have no idea how the first equality between the ranks holds. Just because $L^{(0)} + I$ is invertible, it doesn't mean that $L^{(1)}$ can't be a low-rank matrix whose span gets rotated by $(L^{(0)} + I)^{-1}$ to lie in the nullspace of $L^{(1)}$.
1. [Page 6, paragraph before section 3.4] This whole paragraph is kinda important, but also kinda weak justifications. I don't really know why the tighter inequality makes "it likely that the proposed method with produce better $L^{(t+1)}$" or why the fixed point's non-convexity (but plausible quasiconvexity) would mean that it's possible for it to get stuck in spurious minima.
1. [Page 7] Why should the basic distribution be "unstructured". It's also a Wishart matrix that's concentrated around some (maybe scaled?) identity matrix. It's well studied in random matrix theory. It's gonna have some similar structure to the "Wishart" scheme, no? This is a particular issue for [Page 9], where the you describe the Basic and Wishart schemes as being fundamentally different for the Nottingham dataset.
1. [Page 7] I'm not familiar with the Nottingham dataset. Without an introduction, the paragraphs about Nottingham are incomprehensible. What is the dataset about, what are tracks, what are samples, what are folk tracks in the ABC format, and more questions remains unclear. The baby registry is very well explained, for a point of reference.
1. [Page 9] You say that ADAM is at an advantage for the Nottingham data because it's allowed to be non-monotonic. But it was your choice as authors to force the fixed-point algorithm to be non-monotonic by setting $a=1$. Why pick this values of $a$ if the monotonicity hurts you?
1. [Page 10] The benefit of the proximity term is not clear at all. What do you think will be the upside of using such an algorithm?

---

Typos and minor recommended edits.
1. [Page 1] Replace "Suppose A and B" with "If A and B", and replace "ground set, P(A \cup B)" with "ground set, then P(A \cup B)"
1. [Page 2] Replace "virtual" with "hypothetical"
1. [Page 2] Replace "intensities" with "density" (unless this is a technical term I'm unfamiliar with)
1. [Page 3] Replace "ground set have" with "ground set has"
1. [Page 3] Replace "MLE, that is," with "MLE. That is,"
1. [Page 3] Replace "is a submatrix" with "is the submatrix"
1. [Page 4] Replace "multiplying the both" with "multiplying both"
1. [Page 5] Write $L$ and $\varepsilon$ as inputs to the algorithm... because they are inputs. Otherwise, it's completely unclear how to initialize these parameters until well later (like well into the experiments section).
1. [Page 5] Just before section 3.3, describe some idea on how to initialize L. It's sufficient to say you'll discuss it in the experiments section, but at least acknowledge it.
1. [Page 6] Be more rigorous in the statement "in the neighborhood". Specifically, say the order of quantification. I think it's something like "for each L, there exists a small enough neighborhood around L such that every M with eigenvalues between -delta and +delta has g(L+M) > h(L+M)", or something like that.
1. [Page 6] In Section 3.4, include the number of iterations in the big-Oh, or describe the big-Oh as the cost-per-iteration, rather than the cost of the method.
1. [Page 7] Capitalize the section 4 title
1. [Page 7] Define the fancy U as the uniform distribution.
1. [Page 9] You say that your algorithm achieves better log-likelihood values, but it's by a rather narrow lead. Maybe acknowledge how small that lead is? Up to you, but it felt a bit underwhelming to start by reading that you get better log-likelihoods just to look at the tables and see all methods produce very similar log-likelihoods.
1. [Page 10] "it is not enough" -- be more concrete. What would be enough, and why is this result too weak? What would you hope to be able to prove, for instance?

**Strengths And Weaknesses:**

I like the vibe of the paper, but I have some qualms about the quality of the evidence supporting their proposed algorithm. This leads me to a borderline review of the paper.

At a high level:
- The principle behind the algorithm is nice and simple.
- The theoretical proofs in the paper are too short and hard to follow. I'm not completely convinced of their correctness, despite their brevity.
- Their algorithm computes a lower bound for the likelihood at each iteration. Their core theoretical result claiming that the proposed algorithm outperforms the prior work claims that their computational lower bound is tighter than the lower bound used in a prior work's algorithm. However, this claim only holds in a small neighborhood of the current iterate, and their algorithm may look outside this neighborhood at each iteration. That is, this claim seems a bit mismatched for the algorithm.
- Also, it's not clear to me, even without worrying about this neighborhood, why a tighter lower bound here would translate to a better algorithm.
- The algorithm is very straightforward and easy to implement (top of page 5)
- The experiments show the proposed algorithm often outperforms the existing algorithms.

My overall take is that, if I can be assuaged of my technical concerns about the proof, then I'm inclined to accept the paper as a good fit for TMLR.

---

Now let's get into the weeds of it.

### On the Theoretical Results

The proposed algorithm is based on taking the log-likelihood function (equation 1 on page 3) and taking a taylor approximation of both terms in the sum, producing a lower bound on the actual log-likelihood. Equation 3 on page 3 shows one of these lower bounds, done by Taylor approximation. This approximation is fine, but the approximation for the other term (equations 5 and 6 on page 4), are completely opaque to me. I don't understand how the proof works. _I sort of softly expect there to be an error here even, since the Taylor approximation is used to give both an upper and lower bound on the log determinant across pages 3 and 4, so I'm not clear how that works_.

The math in the paper often skips showing their work, and there are other important lines in the math that I don't follow. I also don't understand how Corollary 3.1 on page 5 is proven, for instance. I'm not a big fan of that, though their claims are not very strong, so I wouldn't be surprised if the math is correct. I don't think they explain it enough though.

The weight of the theoretical results are a bit unclear to me; a bit of a mixed bag. They give three core theoretical guarantees:
- (Prop 3.1 on page 4) Their lower bound on the likelihood is of the right form to be used in an MM style algorithm.
- (Corol 3.1 on page 5) The $L$ matrix and $Q$ matrix in their algorithm have the same rank.
- (Prop 3.3 on page 6) The lower bound in the proposed algorithm is tighter than the lower bound in a prior work, but only in a small enough neighborhood of the current iterative.

The first result is essential for the correctness of their algorithm. The second result is used as evidence suggesting that the algorithm without any regularization would return a degenerate kernel matrix (i.e. with some eigenvalues exactly equal to zero). This is a bit shakey, but I'm okay with it.

The third result... I'm not really sure what it serves to prove. Is there any reason that being tighter in a small enough neighborhood should mean that this nonconvex optimization algorithm should perform better than another nonconvex optimization algorithm? I don't really even see an intuition for why this should be the case. It's certainly not fully argued in the paper. The authors do specifically note that the lower bound ("minorizer") in the prior work is non-convex, which "implies that the [prior work] algorithm is possible [sic] to get trapped in poor stationary points". This point is especially weak of a justification because:
- The proposed algorithm is also a non-convex optimization strategy. It just happens to have one step within it which is convex.
- The other bound being non-convex does not imply that it has spurious minima. The plot in Figure 1(b) on page 6 makes the function look quasiconvex, with no evidence of spurious minima in sight.

### On the Empirical Results

The paper has a nice suite of experiments over a nice set of synthetic and empirical problems. I'm overall pretty happy with it, and it does suggest that the proposed algorithm is at least competitive, and often a good deal better than both the prior work on learning DDPs and the out-of-the-box adam optimizer.

The paper places some emphasis on algorithms can are parameter-free, and easy to work implement as a result. The proposed algorithm is mostly parameter free (it has an epsilon regularization which they recommend setting around machine epsilon; not a big deal). The ADAM optimizer has a good deal of parameters, which they use fairly default values for. The prior work algorithm has a step-size parameter which they set to $a=1$. I take some issue with this.
- They use $a=1$ because the prior work only guarantees that their algorithm converges monotonically when $a=1$. However, the authors of this paper make it clear that the prior work did this in order to improve the empirical performance of their algorithm (page 2, the sentence starting with "In (Mariet & Sra), ..."). The experiments section should acknowledge the empirical techniques used to make the prior work algorithm work better, and it feels odd to not even engage with that in the writing of the experiments section.

Beyond this issue, the experiments are nice, and show that the proposed algorithm typically outperforms the other algorithms considered. They consider a nice spread of experiments, with different initialization techniques. They report parameters well with easy to read plots and tables. It's largely well put together. I've got some pedantry to discuss in the experiments (in the "requested changes" section below), but it's nice and compelling. It especially shows that the proposed algorithm is faster and better performing than the prior work and ADAM algorithms for larger datasets. Often, this isn't a huge speedup relative to the ADAM optimizer, but it's still a win I suppose. It's good stuff.

---

> ### Author Response · Authors · 2023-09-24
> **Response to Reviewer EE9h**
>
> Thank you for the in-depth comments with thorough reading. We summarize our responses and changes based on your requests.
>
> > [Proof of Prop 3.1] Explain the inequalities used more....
>
> [page 4, Eqs (3) and (4); page 4, Prop. 3.1] We clarified the notations that appears in the proof of Prop. 3.1 and how the inequalities are used.
>
> > [Proof of Prop 3.2] Consider showing more work in how you prove equation (8).
>
> [page 5, Prop. 3.2] We first note that the Equation (8) in the previous manuscript has been moved to Equation (11) in the revised manuscript. We made the derivative formulas used in the proof explicit.
>
> > [Proof of Corol 3.1] I have no idea how the first equality between the ranks holds....
>
> In detail, that is proved as follows:
> 1. There exists a full-rank $C \in \mathbb{R}^{N \times N}$ such that $(L^{(0)} + I)^{-1} = CC^\top$.
> 2. $\mathrm{rank}(L^{(1)} (L^{(0)} + I)^{-1} L^{(1)}) = \mathrm{rank}(L^{(1)} CC^\top L^{(1)}) = \mathrm{rank}(L^{(1)}C)$.
> 3. Because $C$ is full-rank, $\mathrm{rank}(L^{(1)} C) = \mathrm{rank}(L^{(1)})$ holds.
>     * This rank equality holds in general (cf. the second proposition in [the blog post (Taboga, 2021)](https://www.statlect.com/matrix-algebra/matrix-product-and-rank))
>
> > [Page 6, paragraph before section 3.4] This whole paragraph is kinda important, but also kinda weak justifications....
>
> We also recognize the weakness of the justification, but it was quite difficult to obtain strong theoretical results. Some complex factors contribute the difficulty, such as (a) there is no existing general theory to compare MM algorithms with different minorizers (majorizers) in the best of our knowledge and (b) because our setting do not satisfies some tractable properties (like strong convexity and/or Lipschitz continuity), known convergence theory of MM cannot be applied (cf. [Lange et al, (2021)](https://arxiv.org/abs/2106.02805)). We believe that the implication from Cor. 3.1 is somewhat informative under the difficulties; tighter lower-bound in Equation (2) means that $f(\theta^{(t+1)})$ is better in the sense of the worst case.
> About the non-convexity of the minorizer of the fixed-point algorithm, perhaps it is quasi-convex as you concern. However, checking the quasi-convexity rigorously is not trivial and is a problem in itself.
>
> > [Page 7] Why should the basic distribution be "unstructured"....
>
> That's not right; even for a scalar $X \sim \mathcal{U}(a, b)$ and $Y = X^2$, we have $p_Y(y) \propto 1/\sqrt{y}$ and $Y$ is not gamma-distributed (which is a special case of Wishart). We note that the naming of the initialization schemes "Wishart" and "Basic" follows the fixed-point's paper (Mariet and Sra, 2015).
>
> > [Page 7] I'm not familiar with the Nottingham dataset....
>
> [page 11, paragraph 3] We added more information about the dataset.
>
> > [Page 9] You say that ADAM is at an advantage for the Nottingham data because it's allowed to be non-monotonic.
>
> [Section 5] In the updated experiments with the newly developed accelerated scheme for the MM, we compare non-monotonic settings in all the algorithm.
>
> > [Page 10] The benefit of the proximity term is not clear at all. What do you think will be the upside of using such an algorithm?
>
> [Section 4] In the newly introduced acceleration for the MM, we exploit a proximity term by considering a "negative proximity" (but the divergence to define a proximity is different with the concluding section in the previous manuscript which you commented on).
>
> ---
> We took most of the minor requests in the revised manuscript. Here we focus on some noteworthy points.
>
> > [Page 2] Replace "intensities" with "density" (unless this is a technical term I'm unfamiliar with)
>
> Because (joint) intensity is a terminology in the field of point processes, we did not changed this.
>
> > [Page 9] You say that your algorithm achieves better log-likelihood values, but it's by a rather narrow lead....
>
> [page 11, paragraph 6; page 12, paragraph 3] In the discussion about the experimental results, we took (statistical) significance of the final log-likelihoods based on their standard deviations.
>
> > [Page 10] "it is not enough" -- be more concrete....
>
> [page 13, paragraph 4] We added a statement about future prospects and modified an expression.

---

### Author Response · Authors · 2023-09-24
**Revised version has been resubmitted**

Dear Reviewers and Action Editor,

Thank you for your careful reading and thoughtful suggestions. Reading the review comments, we have updated our paper and supplementary material includes scripts. Added contents are colored red in the updated paper.
Our update includes some major changes:
- We have further developed an accelerated algorithm of the proposed MM and added a new section for introducing the acceleration.
  * This development had been motivated from the question by Reviewer ENwe and the concluding section in the original version of the paper, in which a possibility of generalization is stated. Thank you!
- The experimental settings have also been changed for using the newly introduced accelerated algorithm.
- We have found a bug in the Adam implementation in the script and fixed it. This affects choice of the learning rate $\eta$ in the experiments.

Other changes that reflect the Reviewers' requests will be discussed in individual responses.

Respectfully,

the authors

---

> ### Author Response · Authors · 2023-10-06
> **Minor corrections were made**
>
> Dear Reviewers and Action Editor,
>
> We found two minor but unforgivable flaws in the revised manuscript submitted previously.
> We have corrected:
> - the description about pairwise negative correlation (in Introduction)
> - the descriptions about experimental settings with default parameters (in Appendix C)
>   * we have clarified the parameter values
>
> In addition to the previous revised version, the corrections are shown in blue. Sorry for the confusion, but we hope this helps with your reviewing.
>
> Sincerely yours,
>
> the authors

---

### Decision · Action_Editor_e867 · 2023-11-01

**Recommendation:** Accept as is

**Comment:**

This manuscript considers the problem of learning a determinantal point process (DPP) from observations. The setting considered is observing a set of samples from the DPP (which are subsets of elements from a ground set) and we seek to estimate the kernel matrix of the DPP to maximize the likelihood of the observed samples. The authors propose a minorization-maximization algorithm for this estimation problem and prove a lower bound on the likelihood it achieves. The lower bound is tighter than the best-known bound for this problem and appears to perform well in a series of experiments.

The reviewers tend to agree that the work presented in this manuscript is high quality and of interest to the TMLR audience, and the majority of them recommend acceptance.

Throughout the review and discussion period, the reviewers provided feedback and suggestions that the authors have faithfully incorporated into the manuscript. I believe these changes have considerably strengthened the work. I commend both authors and reviewers for their engagement throughout this process.

Although I am recommending that this manuscript be accepted as is, I would like to invite the authors to consider the further refinements recommended by reviewer EE9h in their official recommendation.

**Audience:**

Yes, there is no question that the findings of this paper would be of interest to a subset of the TMLR target audience. Determinantal point processes (DPPs) have received sustained interest from the community over the past decade, and the results presented in this manuscript improve upon the current state-of-the-art in DPP kernel approximation.

**Claims And Evidence:**

The claims made in this submission are supported by clear and convincing evidence that was only improved through interaction with the reviewers.